# SWIIFT v0.10: a numerical model of wave-induced sea ice breakup with an energy criterion

Nicolas Guillaume Alexandre Mokus[1], Véronique Dansereau[1,2], Guillaume Boutin[3], Jean-Pierre Auclair[1,4], and Alexandre Tlili[1,5]

[1]Univ. Grenoble Alpes, Univ. Savoie Mont Blanc, CNRS, IRD, Grenoble INP, ISTerre, Grenoble, France
[2]Univ. Grenoble Alpes, CNRS, IRD, Grenoble INP, IGE, 38000 Grenoble, France
[3]Nansen Environmental and Remote Sensing Center, Bergen, Norway
[4]Now at Department of Chemical and Environmental Engineering, Technical University of Cartagena, 30203 Cartagena, Spain
[5]Now at Université Paris–Saclay, CNRS, CEA, Service de Physique de l'État Condensé, 91191 Gif-sur-Yvette, France

**Correspondence:** Nicolas GA Mokus (mokusn@univ-grenoble-alpes.fr)

**Abstract.** The wave-induced breakup of sea ice contributes to the formation of the marginal ice zone in the polar oceans. Understanding how waves fragment the ice cover into individual ice floes is thus instrumental for accurate numerical simulations of the sea ice extent and its evolution, both for operational and climate research purposes. Yet, there is currently no consensus on the appropriate fracturing criterion, which should constitute the starting point of a physically sound wave–ice model. While fracture by waves is commonly treated within a hydroelastic framework and parametrised with a maximum strain-based criterion, in this study we explore a different, energy-based, approach to fracturing. We introduce SWIIFT (Surface Wave Impact on sea Ice – Fracture Toolkit), a one-dimensional model based on linear plate theory, that can produce time-domain simulations of wave-induced fracture, into which we incorporate this energy fracture criterion. We demonstrate SWIIFT with simple simulations that reproduce existing laboratory experiments of the fracture by waves of an analogue material, allowing qualitative comparisons and validations of the energy fracture criterion. We find that under some wave conditions, identified by a dimensionless wavenumber, corresponding to in situ or laboratory wave-induced fracture, the model does not predict fracture at constant curvature; thereby calling into question the appropriateness of parametrising sea ice fracture with a maximum strain criterion.

## 1 Introduction

In the Arctic, the newly available open ocean areas (Raphael et al., 2025) have exposed sea ice to the effects of stronger and more frequent wave events (Thomson and Rogers, 2014). The remaining sea ice is also overall younger, thinner, more fragile and therefore more likely to be fragmented by winds, ocean currents and waves (Stroeve et al., 2012; Stroeve and Notz, 2018). In unconsolidated or fragmented ice, waves are less attenuated (Collins et al., 2015; Ardhuin et al., 2020) and can therefore propagate further into the consolidated part of the ice cover – the ice pack – and break it to a greater extent, thus enabling a wave–ice positive feedback loop (Thomson, 2022; Horvat, 2022).

This wave-induced breakup results in an assembly of floes with sizes ranging from a few metres to hundreds of metres, defining what is generally referred to as the marginal ice zone (MIZ; see Dumont, 2022, and many others), a region whose dynamics is affected by wave propagation. Fragmented sea ice behaves very differently from the consolidated ice pack. It is more mobile, potentially reaching a free drift state, with ice internal stress no longer resisting motions imparted by winds or currents (Alberello et al., 2020), tides (Watkins et al., 2023), or waves (Auclair et al., 2022; Womack et al., 2022), even at high ice concentration. It is also more sensitive to melt (Horvat et al., 2016; Thomson, 2022), as the ratio of lateral surface (proportional to the perimeter and exposed to the ocean) to top surface (exposed to air) increases when the horizontal extent of a floe diminishes, eventually accelerating the disintegration of smaller floes (Toyota et al., 2025). As a result, the MIZ response to storms can result in quick and large sea ice losses (Smith et al., 2018; Blanchard-Wrigglesworth et al., 2022; Cavallo et al., 2025), which could amplify the observed sea ice decline (Asplin et al., 2012; Thomson and Rogers, 2014). Concomitantly, high-frequency sea ice extent variability is missing in state-of-the-art climate models (Blanchard-Wrigglesworth et al., 2021), in which sea ice fragmentation by waves is not accounted for. This suggests an improved representation of the MIZ in sea ice models is essential to deliver accurate predictions of the sea ice evolution, over both short-term and climate time scales. However, it remains challenging as the physical processes controlling sea ice breakup are still largely unascertained, or rest on hypotheses that are not fully backed up by observations.

The first step in this model development should be the identification of a fracture or disintegration criterion, allowing to determine under which wave forcing (amplitude, wavelength, spectral distribution) and ice conditions (thickness, mechanical stiffness) the ice will fragment into floes. To our knowledge, there is actually neither complete physical evidence nor clear consensus within the sea ice community on this criterion. To our knowledge again, all the current wave breakup modelling approaches are based on *local* flexural stress or strain reaching a prescribed critical value, or threshold. Behind this viewpoint is the consideration that maximum deformation will either occur at the crests and troughs of waves (for example, Dumont et al., 2011) or at the wave front (Tkacheva, 2001). Voermans et al. (2020) extended this formalism by combining a strain threshold with wave characteristics into a dimensionless quantity, the value of which separates breakup from non-breakup. The universality of this approach was later called into question (Passerotti et al., 2022). When modelling individual floes, any *local* threshold is however susceptible to be exceeded over large spans of the floes, which makes super-parametrisations necessary. The location of maximum strain or stress is often considered for the fracture location (Dumont et al., 2011; Williams et al., 2017; Montiel and Squire, 2017; Mokus and Montiel, 2022), but other methods, such as computing the strain between successive wave crests and troughs only have been used (Horvat and Tziperman, 2015).

These local threshold-based criteria are consistent with (and usually come hand in hand with) an hydroelastic representation of the wave–ice system, on which a large fraction of the modelling research on wave–ice interaction lies (Squire, 2020). Wave-induced sea ice fracture has thus naturally been considered through this lens (for example, Fox and Squire, 1991; Montiel and Squire, 2017; Zhang and Zhao, 2021; Mokus and Montiel, 2022); even though more novel and computationally involved approaches exist (Herman, 2017; Ren et al., 2021; He et al., 2022). In the hydroelastic framework, the ice is assimilated to an elastic plate that is thin enough for the variations in the buoyancy forces acting on it to be negligible, and that therefore conforms exactly to the shape of the ice–ocean interface. When associated with a critical strain fracturing criterion, this framework has

shown agreement with observations (Kohout et al., 2016; Voermans et al., 2020). It has also allowed wave and floe-resolving numerical simulations to generate steady-state floe size distributions (Kohout and Meylan, 2008; Horvat and Tziperman, 2015; Mokus and Montiel, 2022), and has therefore percolated into coupled global sea ice models (Roach et al., 2019; Bateson et al., 2020; Yang et al., 2024).

The current contribution digs into the question of the criterion for the fracturing of consolidated sea ice by waves, that is, flexural brittle failure. In this, we are motivated by recent laboratory results investigating the response of an ice analogue material to wave forcing (Auvity et al., 2025). In particular, these authors highlighted that the curvature at which their material broke is not constant, but depends monotonically on the applied wavelength. The spread in reported sea ice critical strains (Kohout and Meylan, 2008; Voermans et al., 2020) could thus be an artefact hiding such a relationship. In this context, we
investigate an approach based on a model of fracture propagation in elastic solids (Griffith, 1921) which is common in the field of fracture mechanics. It opposes the energetic cost of creating new surfaces to the elastic energy stored in a material. The resulting energy-based fracture criterion includes the effect of bending deformation as it depends on the associated elastic deformation energy, but is non-local as it is integrated over the length of the deformed ice floe. It leads to a unique solution. Since the original work of Griffith (1921), this model has been updated (Francfort and Marigo, 1998; Francfort, 2021) and built
upon specifically for application to sea ice (Mulmule and Dempsey, 1997; Balasoiu, 2020; Ren et al., 2021). Measurements of sea ice fracture toughness, which can be linked to the energetic cost of fracturing, have been compiled (Dempsey, 1991; Schulson and Duval, 2009) and an extensive body of work also exists on freshwater ice (for example, Gharamti et al., 2021a, b).

With the intent on focusing on the wave-induced deformation and resulting fracturing of brittle ice, we have implemented this energy-based criterion in a framework that differs from the hydroelastic representation in that the ice is not assumed to
conform to the water surface, but freely deforms within the wave field as a result of the local buoyancy and gravity forces. We neglect other processes affecting the seasonal ice zone (Roach et al., 2025, SIZ; see), such as thermodynamics; in particular, we do not handle ice formation within a wave field, and we restrict our study to the case of brittle fracture, excluding the disintegration of a more granular material (dislocation of melting or forming ice). The resulting simple, yet versatile, 1D model also accommodates a strain-based fracture criterion. It thereby allows investigating the effect of using either criterion on the
occurrence of the fracture and, eventually, on the extent of a simulated MIZ, and shape of the associated floe size distribution. Importantly, our model can be stepped forward in time, so that we can use it to follow the propagation of a fracture front as a function of time; in contrast to being able to solely recover the final, fractured state (Horvat and Tziperman, 2015; Mokus and Montiel, 2022). We present an illustration of this capability in Sect. 2.6. In the present paper, we exploit this model in another use case, the comparison to laboratory experiments on fracture, conducted on an analogue material to sea ice. We pursue this
comparison with the particular aim of validating-invalidating the applicability of the energy-based fracturing criterion.

The paper is divided as follows: in Sect. 2, we give a general mathematical description of the model, including the treatment of sea ice as an elastic plate, the formulation of the breakup criteria, and the representation of waves. In Sect. 3, we give specific information pertaining to the numerical results we present in Sect. 4, that is, the particular setup of the model in this study. We discuss these results in Sect. 5.

## 2 Floe and fracture model

In light of the objectives motivating our approach, stated in the introduction, we present in Sect. 2.1 the main physical hypotheses made to achieve a simple, versatile, numerically efficient, yet physically sound model. In Sect. 2.2, we detail our approach to deriving floe deformation, used as an input for the fracture parametrisations presented in Sect. 2.3, and forced by waves discussed in Sect. 2.4. We present numerical aspects in Sect. 2.5 and conclude this Section with an example of time simulation in Sect. 2.6.

### 2.1 Main hypotheses

A common assumption behind wave–elastic plate interaction models (for example, Fox and Squire, 1991; Tkacheva, 2001; Mokus and Montiel, 2022) is to consider the plate thin enough for variations of the buoyancy force acting on it to be negligible. The plate is, however, subjected to the fluid pressure acting on its bottom side, and it is assumed that the plate conforms to the fluid motion at all times. Fluid pressure is determined by solving for the fluid flow, typically by assuming a potential flow and harmonic solutions, with the plate exerting a boundary condition on the fluid domain. To develop the model presented herein, we adopt a different approach, motivated by our interest in the ice deformation and fracture, whereas the focus of fluid-centred models has historically been that of wave scattering and attenuation by the plate. The interested reader can found a comparison between the two approaches in Appendix B.

In this study, the ice cover is considered thick enough for the local changes in buoyancy force not to be negligible. We do not explicitly resolve the fluid flow underneath the plate, and impose no condition on the ice–ocean interface. Instead, we solve for the vertical deflection of the ice stemming from the local balance of gravity and buoyancy forces driven by the sea surface displacement. The fluid surface thus acts as a forcing term, which is made aware of the presence of the ice floes only through parametrised attenuation; a consequence is that floes can locally be immersed. While we limit ourselves to the case of linear elasticity and linear wave forcing, our mechanical formulation interpolates between the limits of an elastic floe that conforms perfectly to the wave surface, and of a rigid floe only capable of solid motion, which can therefore be submerged. We quantify this behaviour with the dimensionless wavenumber $kL_D$, that relates the wavenumber of the forcing wave $k$ (formally introduced in Sect. 2.4.1) to the flexural length of the floe $L_D$ (formally introduced in Sect. 3.1). The small $kL_D$ limit (long wave, compliant floe) corresponds to the strain formulation of Dumont et al. (2011), while the large $kL_D$ limit (short wave, rigid floe) corresponds to their stress formulation.

Ice formation and melt are assumed to happen at timescales beyond that of wave-induced fracture, so that they can be neglected. We do not consider the reflection of waves at the ice–water interface (for example, Mokus and Montiel, 2022), nor viscous deformation of the plate, the compression of an array of floes due to wave radiation (for example, Herman, 2018), or any surge motion, and take note that the model would be more complete if the pressure forcing associated with the contact between the ice and the water was explicitly taken into account.

## 2.2 Governing mechanical equations

Our one-dimensional model considers a fluid volume of finite or infinite depth, equipped with a Cartesian coordinate system $(x, z)$ where $z$ is the vertical coordinate oriented upward, as shown in Fig. 1. We assume translational invariance in the second horizontal direction. The domain is populated with floating ice floes of prescribed positions and lengths, which may not overlap. Any part of the domain not covered with ice is deemed to be open water.

As in the work of Meylan et al. (2015), we model floes as elastic plates, and we derive the deformation of the ice cover using the Kirchhoff–Love thin-plate theory. The ice is thus considered homogeneous, isotropic, and transversally loaded by body forces. We assume a constant thickness $h$ along a given floe, although different floes can have different thicknesses. The two forces acting on the ice are buoyancy and gravity. In the case of a fluid at rest (no waves), equating the gravity force per unit area and the buoyancy force per unit area (thus applying Archimedes' principle) allows expressing the draught of a floe, $d$, as

$$\rho_i h \boldsymbol{g} - \rho_w d \boldsymbol{g} = 0 \Leftrightarrow d = \frac{\rho_i}{\rho_w} h, \tag{1}$$

where $\rho_i$ is the density of the ice, $\rho_w$ the density of the ocean water and $\boldsymbol{g}$ the gravitational field.

We now move away from this rest state, and impose a perturbation of the fluid surface $\eta(x)$. Because the propagation speed of elastic waves in sea ice is several orders of magnitude greater than that of surface gravity waves (Moreau et al., 2020a), we consider this perturbation and the resulting floe deformation to be quasi-static. Thus, we only consider one independent variable, the space coordinate $x$, and no explicit time dependency. The vertical displacement $w(x)$ corresponding to this perturbed state is determined by the local balance between gravity and buoyancy. The weight per unit area of the ice is still $\rho_i h \boldsymbol{g}$. However, the height of displaced fluid now corresponds to the difference between the fluid surface $\eta(x)$ and the displaced bottom of the ice floe, $w(x) - d$. Locally, the buoyancy force per unit area is thus $-\rho_i \big[ \eta(x) - \big( w(x) - d \big) \big] \boldsymbol{g}$; this is illustrated in Fig. 1. The floe is then subjected to the resulting body force

$$\boldsymbol{q}(x) = \rho_i h \boldsymbol{g} - \rho_w \big[ \eta(x) - \big( w(x) - d \big) \big] \boldsymbol{g} \tag{2}$$

$$= -\rho_w [\eta(x) - w(x)] \boldsymbol{g}, \tag{3}$$

and projecting $\boldsymbol{q}$ onto the vertical axis gives

$$q(x) = \rho_w g [\eta(x) - w(x)]. \tag{4}$$

Using the bending equation of a loaded plate, we then obtain a differential equation on the deflection of the floe,

$$D \frac{\mathrm{d}^4 w}{\mathrm{d}x^4} = \rho_w g \big[ \eta(x) - w(x) \big], \tag{5}$$

where we take advantage of the simplifying hypotheses made on our geometry. In this equation, we introduced the flexural rigidity $D = \frac{Y h^3}{12(1 - \nu^2)}$, characterising the ability of the plate to resist bending, with $Y$ and $\nu$ the Young's modulus and Poisson's ratio of the plate. We complete Eq. (5) with free-edge boundary conditions, that is, vanishing moment and force at both ends

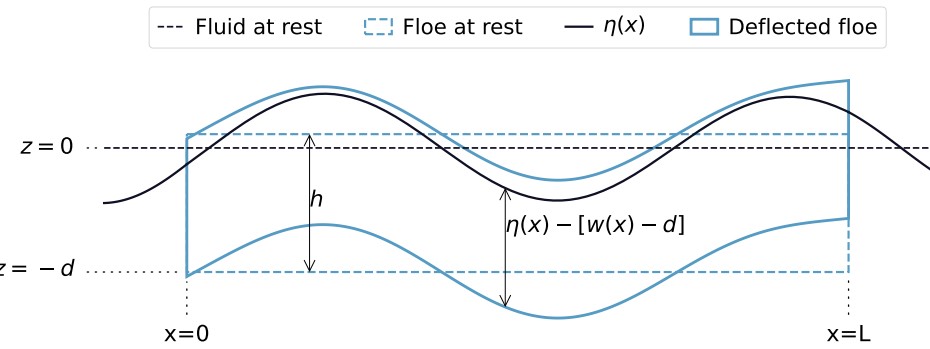

**Figure 1.** Schematic of a deformed ice floe in a wave field. The horizontal dashed line represents the sea surface at rest in the absence of ice, which we use as the reference level. The dashed rectangle represents a floe at rest. A perturbation of the fluid surface (solid line) in the free-surface regions imposes a deflection of the ice floe (solid-lined shape). This corresponds to a model output, with ice thickness $50\,\mathrm{cm}$, floe length $120\,\mathrm{m}$, forcing wave with amplitude $20\,\mathrm{cm}$ and wavelengths $76.4\,\mathrm{m}$ (open water) and $84.3\,\mathrm{m}$ (ice-covered water).

of the ice floe of length $L$. We choose a reference frame local to the floe, where $x = 0$ corresponds to its left edge, so that the complete boundary problem can be written

$$
\begin{cases}
\dfrac{\mathrm{d}^4 w}{\mathrm{d}x^4} = \left(\dfrac{D}{\rho_w g}\right)^{-1} \left[\eta(x) - w(x)\right] & x \in [0, L] \quad\quad\quad (6\mathrm{a})\\[2ex]
\dfrac{\mathrm{d}^2 w}{\mathrm{d}x^2} = 0 & x \in \{0, L\} \quad\quad\quad (6\mathrm{b})\\[2ex]
\dfrac{\mathrm{d}^3 w}{\mathrm{d}x^3} = 0 & x \in \{0, L\}. \quad\quad\quad (6\mathrm{c})
\end{cases}
$$

For prescribed wave conditions and material properties, solving Eq. (6) thus provides the deflection $w$ of the floe.

We focus here on the bending undergone by an elastic plate because of a perturbation of its fluid foundation. We recall that we do not explicitly resolve the fluid motion itself, nor the translational motions imparted to the plate by the fluid. In particular, we thus neglect surge motion, that is, ice drift in the direction of wave propagation.

## 2.3  Fracture

### 2.3.1  Energy criterion

Unlike the prevalent maximum strain formalism commonly used by the sea ice community when modelling wave–ice interactions (for example, Kohout and Meylan, 2008; Dumont et al., 2011; Horvat and Tziperman, 2015; Mokus and Montiel, 2022), we develop a breakup criterion from the framework of fracture mechanics, based on the consideration that in solid, brittle materials, fracture happens to minimise the internal energy associated with deformation (Griffith, 1921). In this framework, the total energy to be minimised is the sum of the elastic energy $E_{\mathrm{el}}$ associated with the deformation (in our case, bending) of the

material, and of the fracture energy $E_{\mathrm{fr}}$ associated with the creation of new surfaces around a crack. This energy decomposition is consistent with mode I fracturing, which in the case of our model translates to vertical fractures due to in-plane tensile stress.

The elastic energy density (per unit length in the transverse horizontal direction, and per unit thickness) stored in a material that is elastic, isotropic, and homogeneous, and stretched only in the longitudinal direction, is

$$W_{\mathrm{el}} = \frac{D}{2h}\kappa^2(x) \qquad (7)$$

with

$$\kappa(x) = \frac{\mathrm{d}^2 w}{\mathrm{d}x^2} \qquad (8)$$

the local linear floe curvature due to the deformation. Equation (7) stems from integrating the density of elastic energy (per unit volume) along the axis normal to the neutral plane of the plate (in our case, the $z$-direction), and takes into account stretching or compression in the directions of the plane (in our case, simply the $x$-direction). By integrating Eq. (7) along the floe, we obtain the surface energy density,

$$E_{\mathrm{el}} = \frac{D}{2h}\int_0^L \kappa^2(x)\,\mathrm{d}x. \qquad (9)$$

The fracture energy density $E_{\mathrm{fr}}$ relates to the energy required to create a new surface. In the case where the only admissible fractures vertically break the ice through its entire thickness, we simplify the formulation from Francfort and Marigo (1998) as

$$E_{\mathrm{fr}} = N_{\mathrm{fr}}G \qquad (10)$$

with $N_{\mathrm{fr}}$ the number of fractures, and $G$ the energy release rate. Again, note that this energy is expressed per unit surface normal to the $x$-direction.

To determine whether a floe breaks, we compare two energy states: that of the unbroken, deformed floe, and a hypothetical state in which this floe has fractured into several fragments. In the former state, the elastic energy, noted $E_{\mathrm{el}}^0$, is that of the deformed floe, as defined in Eq. (9). In the latter state, the total elastic energy, noted $E_{\mathrm{el}}^s$, is the sum of the elastic energies of the individual newly broken floes. If the broken state is – from an energy standpoint – favourable, it should replace the unbroken state. Formally, we look for the finite set of the fracture locations, $\boldsymbol{x}^{\mathbf{fr}} = \{x_j \mid x_j \in (0, L)\}$. This set has size $N_{\mathrm{fr}}$, the number of fractures, dividing the original floe into $N_{\mathrm{fr}} + 1$ fragments. It should minimise the free energy $F$ defined as

$$F(\boldsymbol{x}^{\mathbf{fr}}) = E_{\mathrm{fr}}(N_{\mathrm{fr}}) + E_{\mathrm{el}}^s(\boldsymbol{x}^{\mathbf{fr}}) - E_{\mathrm{el}}^0 \qquad (11)$$

with the additional constraint that for breakup to occur, we must have $F < 0$. In other words, a floe breaks if the elastic energy released by the breakup exceeds the energetic cost of the breakup. If no such set can be found, we conclude that the current deformation of the floe is not sufficient to fracture it. The post-fracture elastic energy expands to

$$E_{\mathrm{el}}^s(\boldsymbol{x}^{\mathbf{fr}}) = \frac{D}{2h}\sum_{j=0}^{N_{\mathrm{fr}}}\int_{x_j}^{x_{j+1}} \kappa_j{}^2(x)\,\mathrm{d}x \qquad (12)$$

with $x_0 = 0$ and $x_{N_{\text{fr}}+1} = L$. The curvatures $\kappa_j$ are obtained from solving Eq. (6) individually for every (at this stage, still hypothetical) fragments.

Equation (11) has an explicit dependency to the number of fractures allowed to happen for a given quasi-static state, that is, at a given time. It suggests that the size of $\boldsymbol{x}^{\text{fr}}$ should be a dimension to the minimisation problem. In practice, when considering travelling waves, floes of reasonable size, and the succession of such quasi-static states, at most one single fracture is admissible at a given timestep, which greatly diminishes the numerical cost of the procedure. In what follows, we will thus use $N_{\text{fr}} = 1$, $\boldsymbol{x}^{\text{fr}} = \{x_1\}$, and we will have

$$E_{\text{el}}^s(x_1) = E_{\text{el}}^<(x_1) + E_{\text{el}}^>(x_1) \tag{13}$$

with $E_{\text{el}}^<(x_1), E_{\text{el}}^>(x_1)$ the elastic energies of the left ($x < x_1$) and right ($x > x_1$) fragments obtained from that single fracture, while the fracture energy reduces to $E_{\text{fr}} = G$. Hence, we look for

$$\begin{cases} x_{\text{fr}} = \underset{x_1 \in (0,L)}{\arg\min}\left(E_{\text{el}}^< + E_{\text{el}}^>\right), & \text{(14a)} \\ F(x_{\text{fr}}) < 0. & \text{(14b)} \end{cases}$$

For given ice and wave conditions, fracture search can thus be conducted in a completely deterministic manner. In practice, we proceed by sampling $E_{\text{el}}^s(x_1)$ regularly on $(0, L)$, ensuring the sampling rate is sufficient to capture its oscillations. We find the set of arguments of the peaks (the local maxima) of this discretised $E_{\text{el}}^s$, which we augment with the bounds $\{0, L\}$ of the domain, to obtain an ordered sequence of at least two coordinates bounding, two by two, local minima of $E_{\text{el}}^s$. We conduct local minimisation between the bounds using Brent's method (Virtanen et al., 2020). Finally, the smallest of these minima is validated against Eq. (14b). If this condition is verified, its argument is the fracture location $x_{\text{fr}}$. These steps are summarised in Fig. 2, and a fracture search is illustrated in Fig. 3. Note that in this case, the asymmetry of the total energy profile comes from differences in wave phase at the edges of the floe, and wave attenuation by the ice cover, discussed in Sect. 2.4.

### 2.3.2 Strain criterion

To allow future comparisons, we additionally implement a conventional strain criterion for fracture. Under that formulation, the floe is allowed to fracture if the bending strain $\varepsilon$ locally exceeds a prescribed critical strain $\varepsilon_{\text{cr}}$, that is if

$$\exists x \mid |\varepsilon(x)| > \varepsilon_{\text{cr}} \tag{15}$$

with

$$\varepsilon(x) = -\frac{h}{2}\frac{\mathrm{d}^2 w}{\mathrm{d}x^2} \tag{16}$$

the maximum (when taking the absolute value) bending strain, here defined as evaluated at the top of the floe.

Typically, if Eq. (15) holds anywhere, it holds on continuous intervals along the floe. We illustrate this in Fig. 4. A second criterion must then be chosen to constrain the fracture. Herein, we arbitrarily choose to consider the global strain extremum

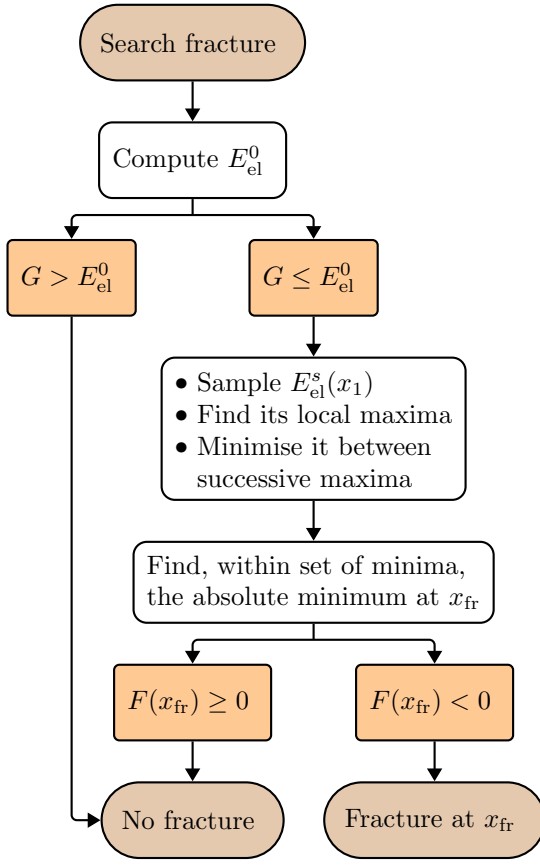

**Figure 2.** Algorithmic steps behind a fracture search.

(single fracture). We thus have

$$
\begin{cases}
x_{\text{fr}} = \underset{x \in (0,L)}{\arg\max} |\varepsilon(x)|, & \text{(17a)} \\
|\varepsilon(x_{\text{fr}})| > \varepsilon_{\text{cr}}. & \text{(17b)}
\end{cases}
$$

This criterion can be straightforwardly extended to the case of multiple fracture by considering all the local extrema exceeding the critical value.

### 2.3.3 Values of fracture parameters

Our two fracture parametrisations rely on two different parameters: the energy release rate $G$ (energy criterion) or the critical strain $\varepsilon_{\text{cr}}$ (strain criterion). These parameters have to be measured or estimated from sea ice samples. Sea ice properties can vary greatly based on its history and environmental conditions, such as temperature, brine fraction, or past loading rate.

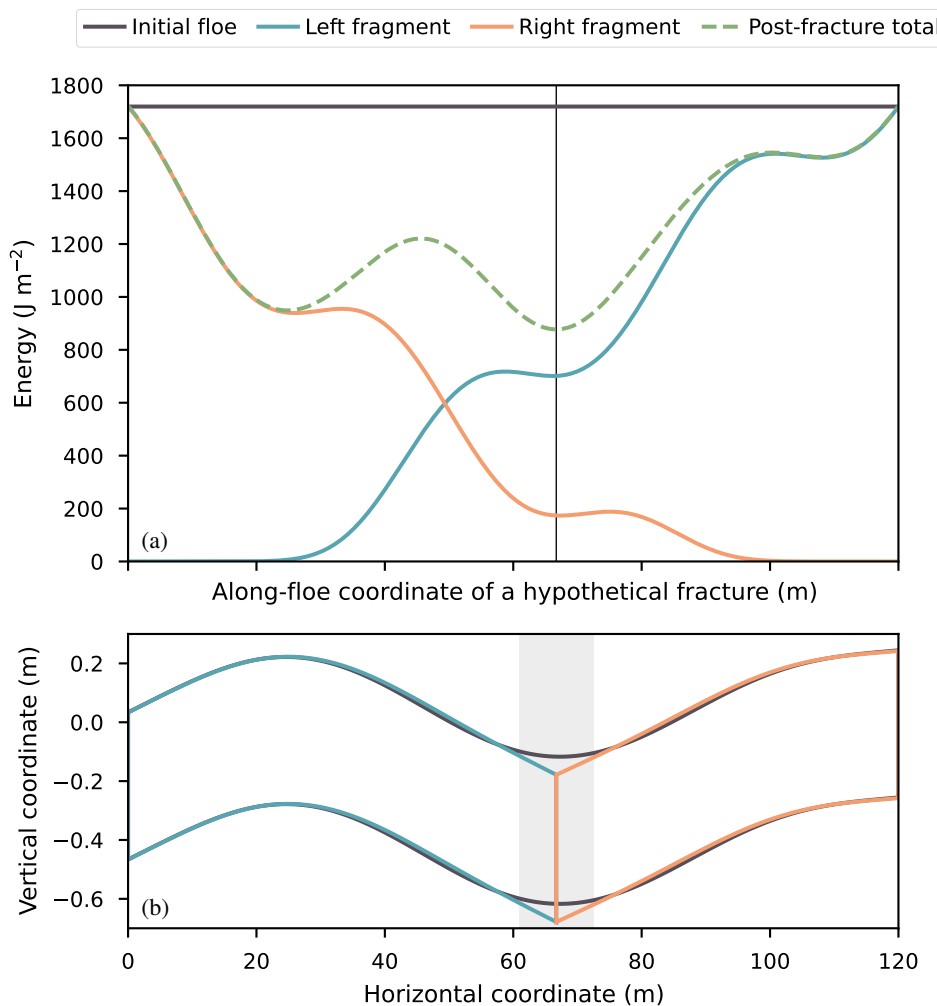

**Figure 3.** Illustration of a fracture search, from a situation corresponding to Fig. 1. In (a), contributions to the system's elastic energy change according to the coordinate of the fracture, and the total energy (that includes the fracture energy) is compared to the energy of the initial, unfractured floe in order to determine whether fracture should occur. The vertical line locates the global minimum, $x_{\text{fr}}$, of total energy which is, according to our model, where the floe should break. In (b), representation of the deformed floe, before and after fracture at $x_{\text{fr}}$. The shaded rectangle represents the energy relaxation length, as defined in Eq. (27), centred on $x_{\text{fr}}$.

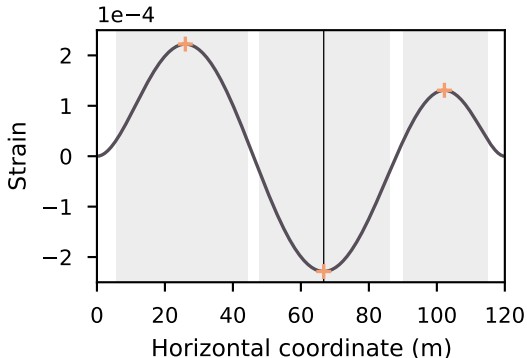

**Figure 4.** Strain-based fracture parametrisation, for a situation corresponding to Fig. 1. The line represents the maximum bending strain along the floe. The shaded vertical strips indicate where the strain exceeds, in absolute value, a typical critical strain of $\varepsilon_{\mathrm{cr}} = 3 \times 10^{-5}$. The crosses indicate local extrema, and the vertical line the global extrema.

Nevertheless, $G$ and $\varepsilon_{\mathrm{cr}}$ are physical quantities that can be measured, and are not completely free parameters. For a detailed compilation of sea ice property measurements, we refer the reader to Timco and Weeks (2010).

Additionally, ice strength depends on the direction of the applied stress. We are here solely interested in failure from bending (mode I, or opening mode; see Saddier et al. (2024)), which is compatible with Griffith's model of fracture as well as with wave action. In particular, in the plane strain approximation, $\varepsilon_{\mathrm{cr}}$ can be related to the flexural strength $\sigma_{\mathrm{f}}$ so that $\varepsilon_{\mathrm{cr}} = \frac{1-\nu^2}{Y}\sigma_{\mathrm{f}}$, and $G$ to the fracture toughness $K_{1\mathrm{c}}$ so that $G = \frac{1-\nu^2}{Y}K_{1\mathrm{c}}^2$. Schulson and Duval (2009) compile values of $K_{1\mathrm{c}}$ in the range 75 to $150\,\mathrm{kPa\,m}^{\frac{1}{2}}$. Wei and Dai (2021) measured values down to $26\,\mathrm{kPa\,m}^{\frac{1}{2}}$ for floating samples in the lab, a reduction that could be attributed to temperature or size effects (Dempsey et al., 1999). Reported values of $\varepsilon_{\mathrm{cr}}$ are typically in the range of $1 \times 10^{-5}$ to $1 \times 10^{-4}$ (Kohout and Meylan, 2008), even though larger value (on the order of $1 \times 10^{-3}$) have been reported for lab-grown, saline ice (Herman et al., 2018). Sea ice is subject to fatigue, and repeated cyclic loading was shown to lower its apparent flexural strength (Langhorne et al., 1998).

## 2.4 Forcing waves

The main focus of this study being ice deformation and fracture, the wave component of the model is kept relatively simple. To align with the linearity hypothesis made on elastic plates, we only consider linear plane waves. This is also in line with previous studies (for instance, Kohout and Meylan, 2008; Dumont et al., 2011; Horvat and Tziperman, 2015; Mokus and Montiel, 2022).

### 2.4.1 Dispersion relations

For a prescribe angular frequency $\omega$, we derive wavenumbers $k$ from the dispersion relations

$$\frac{\omega^2}{g} = k\tanh(kH) \tag{18}$$

in the open-water parts of the domain, and

$$\frac{\omega^2}{g} = \left(\frac{D}{\rho_w g}k^4 + 1 - \frac{\omega^2}{g}d\right)k\tanh\big(k(H-d)\big) \tag{19}$$

in the ice-covered parts. We use the single symbol $k$ for brevity, and the appropriate dispersion relation should be understood from context. In the right-hand side of Eq. 19, the term $\frac{D}{\rho_w g}k^4$ corresponds to the elastic response of the ice cover, while the term $\frac{\omega^2}{g}d$ corresponds to its mass-loading response. Whether the former has a significant contribution to the dispersion relation when the ice is heavily fragmented is debated (Sutherland and Dumont, 2018; Dumas-Lefebvre and Dumont, 2023). As it can easily be turned off, we include it here for completeness.

We note that this relation dispersion can be derived by considering the bending of a plate conforming to a fluid surface excited by harmonic waves. This can therefore be seen as a soft coupling of the fluid to the plate. The dispersion relation of plate excited by harmonic waves, without a fluid foundation, would otherwise be $\omega^2 = \frac{D}{\rho_i h}k^4$.

### 2.4.2 Sea state

For any given floe in the domain, a linear monochromatic wave can be parametrised with two complex variables, amplitude $\hat{a}$ and wavenumber $\hat{k}$. The modulus $a = |\hat{a}|$ denotes the amplitude of the wave at the left edge of the floe, while the argument $\phi = \text{Ang}\,\hat{a}$ denotes the phase of that wave mode at the left edge of the floe. The real part of the wave number, $k = \text{Re}\,k$, describes wave propagation while its imaginary part $\alpha = \text{Im}\,\hat{k}$ describes the spatial rate of attenuation in the direction of propagation. Following Sutherland et al. (2019), we implement a parametrisation with attenuation linear in ice thickness and quadratic in wavenumber, so that

$$\alpha = \frac{1}{4}hk^2. \tag{20}$$

Other parametric attenuations can easily be added to the current framework; either directly to SWIIFT's codebase[1] or by a user at run time. Attenuation can also be turned off altogether.

A linear polychromatic plane wave can be defined by superposition. The wave state is then

$$\eta(x) = \sum_j \text{Im}[\hat{a}_j \exp(\hat{k}_j x)] \tag{21}$$

where the subscript $j$ denotes spectral modes. The modal amplitudes can be derived from any spectral density $S(\omega)$, using the relationship (Horvat and Tziperman, 2015)

$$\frac{1}{2}a_j^2 = S(\omega_j)\Delta\omega_j \tag{22}$$

where $\Delta\omega_j$ is the width of the angular frequency bin corresponding to the amplitude $a_j$.

---

[1]A parametrisation derived from Yu et al. (2022) was added to a later version.

### 2.4.3 Wave propagation over a finite distance

To allow for the advection of a developing sea into the ice-covered domain, we apply a semi-Gaussian kernel to the sea state.
We implement this modification to avoid the non-realistic situation of a fully developed sea appearing under a potentially kilometre-wide MIZ. Therefore, Eq. (21) is modified into

$$\eta(x) = \sum_j \text{Im}\big[\hat{a}_j \exp(\hat{k}_j x)\big] K_j(x) \tag{23}$$

with

$$K_j(x) = \begin{cases} \exp\left(-\frac{(x-\mu_j)^2}{2\sigma_j{}^2}\right) & x \geq \mu_j, \\ 1 & x \leq \mu_j. \end{cases} \tag{24}$$

To each wave mode, we associate a coordinate $\mu_j$. The wave is considered fully developed in the half-plane left of that coordinate. The parameters $\sigma_j$ control the width of the transition between a fully developed wave, and a near-rest state (as the wave envelop of a given mode is reduced to about $0.01\,\%$ of its maximum at $x = \mu_j + 3\sigma_j$).

## 2.5 Numerical scheme

We have so far presented our framework for modelling fracture in a quasi-static state. Here, we give more details on how we iterate from a quasi-static state to the next, and summarise the steps leading to evaluating ice floe fractures.

### 2.5.1 Wave propagation–attenuation

Let $\tau$ be a model timestep. Each wave mode in Eq. (23) propagates at a phase speed

$$c_j = \frac{\omega_j}{k_j}. \tag{25}$$

Between time $t_n$ and time $t_{n+1}$, the phase $\phi_j$ of mode $j$ increases by $-\omega_j\tau$. The limit $\mu_j$ of the fully developed wave advances of a distance $c_j\tau$. Therefore, we iterate in time by updating the values of our $\hat{a}_j$ and $\mu_j$ by these quantities. A new quasi-static wave profile $\eta(x)$ can then be computed across the domain.

### 2.5.2 Sea ice deformation and fracture

Once the sea surface has been computed, the resulting sea ice deflection $w(x)$ is computed for all individual floes, which are scanned for possible fractures. For a given floe, if $\mu_j > L \;\forall j$ (that is, the wave acting on the floe is fully grown), the deflection can be determined analytically, as developed in Appendix A. Otherwise, we obtain a solution to Eq. (6) with a numerical solver (Virtanen et al., 2020). In turn, the deflection is used to compute the curvature. Depending on the chosen fracture mechanism (energy-based or strain-based), the floes are considered for breakup, as described in Sect. 2.3.1 and 2.3.2.

If using the energy criterion, we evaluate the post-fracture total energy along the discretised floe. We use a peak detection algorithm (Virtanen et al., 2020) to separate intervals of convex free energy (as can be identified in Fig. 3a), onto which Eq. (13)

is minimised. If the global minimum among these local minima satisfies Eq. (14b), fracture occurs; these steps are summarised in Fig. 2. If using the strain criterion, we evaluate the bending strain along the discretised floe. Again, a peak detection algorithm is run on $-\varepsilon^2(x)$ to detect convex intervals, and we conduct local minimisation on these, which is equivalent to maximising $|\varepsilon(x)|$. If the global minimum satisfies Eq. (17b), fracture occurs.

The input necessary for both parametrisations is thus the floe curvature. In Sect. 2.2 and Sect. 2.4, we merely suggest a simple mechanical model to infer this curvature from wave forcing. Other 1D models outputting the curvature field, or actual curvature measurements, can be substituted without having to alter the fracture formalism presented in Sect. 2.3. However, for the fracture parametrisation to be sensible, it is necessary that the mechanical model can be stepped forward in small time increments. Thus, we choose here not to rely on harmonic solutions to the bending problem, such as in Mokus and Montiel (2022), as these rely on the hypothesis that a steady state has been reached in the whole fluid domain. We do so at the cost of neglecting floe bending inertia and relaxing constraints on the fluid itself. In particular, wave scattering induced by different boundary conditions imposed on the fluid when transitioning between open water and ice-covered water regions is not accounted for. A comparison between the two types of solution can be found in Appendix B. We find minor differences in terms of floe curvature (impacting the strain parametrisation) and resulting elastic energy (impacting the energy parametrisation). We compute the ratio of elastic energy ($E_{\mathrm{el}}$, defined in Eq. (9)), derived from the scattering model, to that same energy derived from SWIIFT, for the case of a polychromatic forcing. The elastic energy is, generally ($77.5\,\%$ of the ensemble considered), overestimated by SWIIFT; but the distribution of these ratios being skewed, the two models yield, on average, a similar value (mean of the ensemble considered: $1.02$, geometric mean: $0.69$). Additionally, we do not find these ratios to depend on model parameters such as ice thickness or floe length. We thus conclude that even though differences exist between the two solutions, they are less meaningful than random fluctuations of the wave state.

### 2.5.3 Timestep selection

Care must be taken when selecting a model timestep, $\tau$. The theoretical upper limit for crack propagation in an elastic, isotropic, and homogeneous material is set by the speed of Rayleigh waves, $c_{\mathrm{R}} = \sqrt{\frac{Y}{2\rho_i(1+\nu)}}$. Using $Y = 3.8\,\mathrm{GPa}$ and $\nu = 0.33$, values of the Young's modulus and Poisson's ratio estimated in situ for sea ice (Moreau et al., 2020b), this speed is on the order of $c_{\mathrm{R}} \approx 1250\,\mathrm{m\,s^{-1}}$. As we consider cracks to instantaneously fracture floes through their thickness, we must have $\tau > \frac{h}{c_{\mathrm{R}}}$. For $1\,\mathrm{m}$ thick ice, it corresponds to $\tau > 0.8\,\mathrm{ms}$. Fractures propagate faster in stiffer ice, and increasing the Young's modulus (or reducing the thickness) would lower this bound, which must be considered on a case-by-case basis.

However, we also want to keep the timestep small enough that we can detect fractures as soon as it is possible for them to occur, as delaying the onset of a fracture may affect the length of the resulting floes. Therefore, we aim to keep the ratio of the progression of the wave front to the wave amplitude small. In the monochromatic case, with phase speed $c$, it translates to having $\frac{\tau c}{a} < r \Leftrightarrow \tau < r\frac{ak}{\omega}$, with $r < 1$. Setting $r = \frac{1}{5}$ ensures sufficient convergence. An analogous relationship can be derived for polychromatic cases, by substituting the amplitude by the significant wave height $H_{\mathrm{S}}$ of the spectrum, and the phase speed $c$ by the maximum phase speed of within the sampled spectrum, so that $\tau < r\frac{H_{\mathrm{S}}}{\max_j c_j}$.

## 2.6 Example of time simulation

The study of floe size distributions in relation to ice or wave parameters is out of the scope of this study. However, as an illustration of the capabilities of SWIIFT, we present in this Section the result of a single simulation.

We initialise the domain with a single floe of length $L = 600\,\mathrm{m}$, thickness $h = 50\,\mathrm{cm}$, Young's modulus $Y = 4\,\mathrm{GPa}$, and Poisson's ratio $\nu = 0.3$. We choose to parametrise fracture with the energy criterion, and set the fracture toughness to $K_{1\mathrm{c}} = 100\,\mathrm{kPa\,m}^{\frac{1}{2}}$, which together with the other mechanical parameters corresponds to an energy release rate $G = 2.275\,\mathrm{J\,m}^{-2}$.

This floe is forced with waves issued from a (one-parameter) Pierson–Moskowitz spectrum, with significant wave height $H_\mathrm{S} = 0.5\,\mathrm{m}$ (corresponding to a peak period of $3.84\,\mathrm{s}$), truncated to the period interval $T \in 1$ to $15\,\mathrm{s}$. We discretise this spectrum onto 50 linearly spaced frequency bins, and thus obtain 50 tuples of amplitudes and wavelengths, which we complete with 50 initial phases sampled from a uniform distribution in 0 to $2\pi\,\mathrm{rad}$. Spatial attenuation is parametrised as defined in Equation (20). We initialise the growth kernels $K_j$ with standard deviations $\sigma_j$ equal to the wavelengths, and means $\mu_j$ equal to three respective standard deviations upstream from the floe. At time $t = 0\,\mathrm{s}$, the magnitude of the surface perturbation at the left edge of the floe, issued from wave superposition as defined in Equation (23), is about $0.2\,\mathrm{mm}$.

We set the timestep $\tau = \frac{1}{5}\frac{H_\mathrm{S}}{\max_j c_j} = 8.58\,\mathrm{ms}$. We run the simulation for $120\,\mathrm{s}$; the first fracture occurs at $t = 9.097\,\mathrm{s}$, the last one at $t = 105.497\,\mathrm{s}$. We present results of this fracture experiment in Fig. 5, showing a snapshot of the simulated fluid and ice displacement along with the evolving number and lengths of the simulated fragments. A video of the simulation is available as supplementary material (Mokus, 2025a).

## 3 Numerical experiment

To evaluate the capabilities of our model, in particular, validate the energy-based fracturing approach and highlight the difference between energy and strain criteria, we choose to replicate breakup experiments conducted at the laboratory scale on a material that served as an analogue for solid, cohesive ice (Auvity et al., 2025). These focused on quantifying the onset of breakup, by progressively increasing the amplitude of a forcing stationary wave, at different frequencies. The experiment setup was as follows: a water tank of length $80\,\mathrm{cm}$ and depth $11\,\mathrm{cm}$ was covered with a brittle layer of varnish, with thickness on the order of $100\,\mu\mathrm{m}$. The layer was detached from the walls of the tank prior to the experiment. Stationary surface waves were generated with a wave maker. A one-dimensional profilometry system and image-processing method were used to extract the wave properties (frequency, amplitude, wavenumber) and determine when fracture occurred. This work is similar to that of Saddier et al. (2024), who also conducted wave-induced fracture experiments on an analogue material at the laboratory scale, under stationary but also progressive forcing. However, in their experiment, the material is a granular raft hold together by capillary forces more than a continuous solid, and breaks because of viscous stress rather than because of bending stress. The former is directly relevant for representing the disintegration of an already fragmented and granular sea ice, that has already been broken up or that is in a consolidation phase (transition from frazil to grease ice). As our work focuses on the fracturing of a solid and cohesive ice cover that we treat as a continuous elastic medium, rather than on the disintegration of a granular ice of low cohesion, we favoured the work of Auvity et al. for our comparisons.

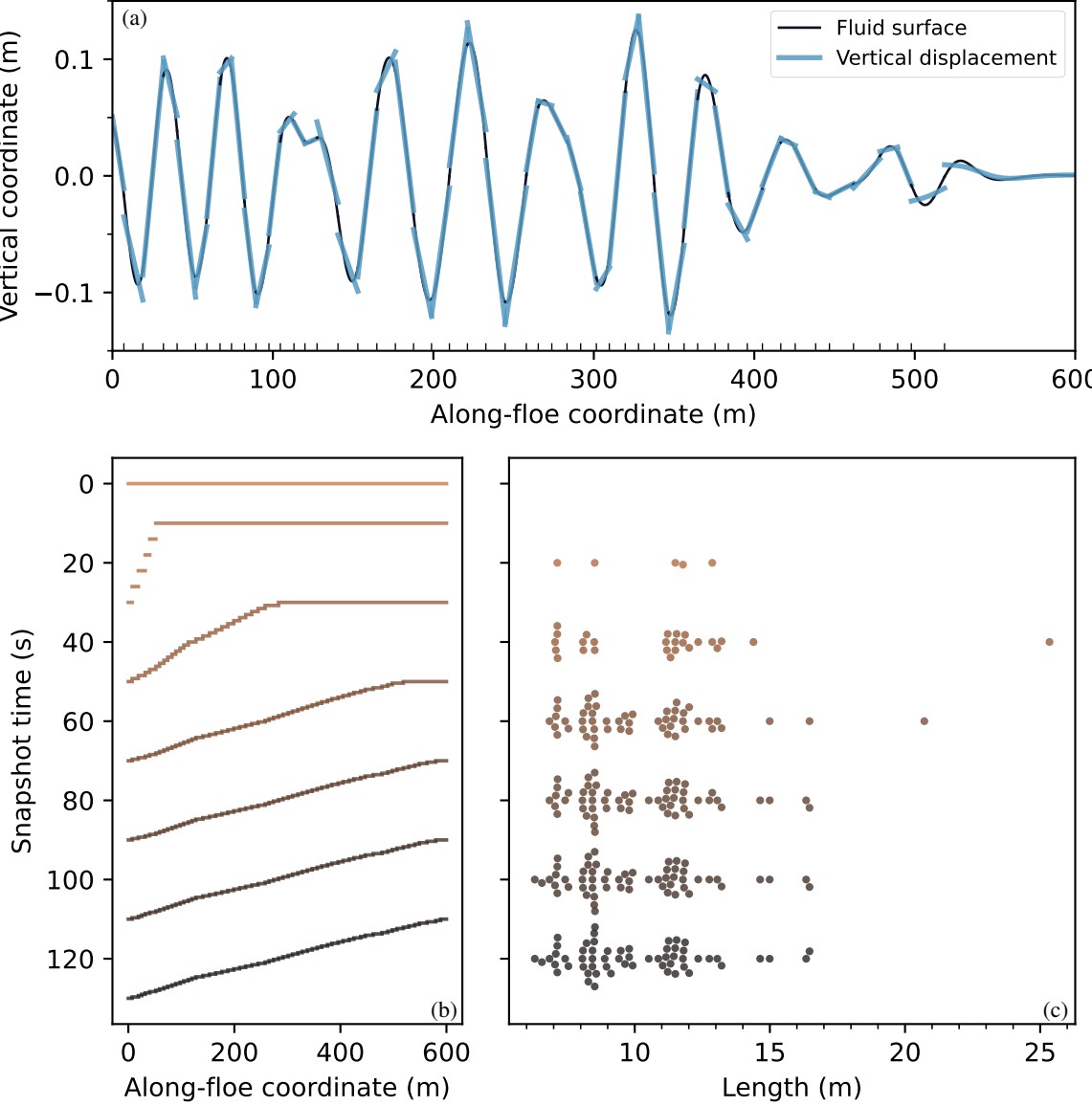

**Figure 5.** Snapshots of a fracture experiment. In 5a, view of the domain at $t = 60\,\mathrm{s}$. The continuous, dark line represents the fluid surface ($\eta(x)$), and the discontinuous, lighter lines the vertical displacements ($w(x)$) of individual floes. The marks along the bottom spine indicate the boundaries between fragments; the last $80\,\mathrm{m}$, at the right of the domain, have not yet been affected by the waves. Note that the vertical scale is greatly exaggerated: the aspect ratio of the graph, in physical units, is $5 \times 10^{-4}$. Because of the thickness of the lines, some floes appear to overlap, they actually do not. In 5b, horizontal bars show the extent of individual floes. The height of the bars indicates the order of the floe in the array, and each group of bars, or "stair", corresponds to a snapshot. The time of the snapshots are indicated on the y-axis, and darker colours correspond to later times. In 5c, we show size distributions as swarmplots, omitting the rightmost fragment. Each dot corresponds to a length as indicated by the x-axis, and within a group, the y-axis only serves to separate dots. Vertical clusters thus indicate a concentration of observations around the corresponding length. From $t = 0\,\mathrm{s}$ to $120\,\mathrm{s}$, there are respectively 1, 6, 27, 52, 58, 59 and 60 fragments.

As we aim to replicate this experiment, in what follows, we will be using a standing wave forcing, and turn off any attenuation. We use our model to determine, for prescribed wavenumbers and material properties inherited from these laboratory experiments (listed in Table 1), the critical amplitude $a_{cr}$ at which the material starts to fracture. We do so using our energy formulation. Our model being linear, the amplitude directly controls the deflection of the plate, and thus, its curvature and resulting elastic energy. It is therefore an intuitive quantity to control the outputs of the model, as well as a quantity that was measured experimentally.

The critical amplitude can then be related to a critical curvature $\kappa_{cr}$ by evaluating Eq. (8) at the coordinate of the fracture. In turn, $\kappa_{cr}$ can be used to derive a critical strain, using Eq. (16). We do not run separate experiments based on a strain criterion, as per the results of these authors, a critical strain independent of the wave forcing does not seem to exist for this material and therefore cannot be prescribed in numerical experiments. Even so, the results of energy-based simulations allow us to draw conclusions on the relevance of this type of criterion. These are discussed in Sect. 5.

## 3.1 Length scales

To replicate the experimental protocol of Auvity et al. (2025), we consider only monochromatic stationary forcings, so that $\eta(x) = a\sin(k_n x)$ with $k_n = \frac{n\pi}{L}$. The symbol $L$ represents both the length of the plate, and of the domain. The positive integer $n$ is the harmonic number. The wave tank we simulate is short enough for attenuation to be considered insignificant.

We define two additional lengths: the flexural length

$$L_D = \left(\frac{D}{\rho_w g}\right)^{1/4},$$
(26)

and the relaxation length

$$L_\kappa = \frac{\int_0^{x_{fr}} (x_{fr} - x)\left[\kappa(x) - \kappa^<(x)\right]^2 \mathrm{d}x}{\int_0^{x_{fr}} \left[\kappa(x) - \kappa^<(x)\right]^2 \mathrm{d}x} + \frac{\int_{x_{fr}}^{L} (x - x_{fr})\left[\kappa(x) - \kappa^>(x)\right]^2 \mathrm{d}x}{\int_{x_{fr}}^{L} \left[\kappa(x) - \kappa^>(x)\right]^2 \mathrm{d}x}.$$
(27)

The former is a natural length scale of our system, appearing in the bending equation (Eq. (6a)), and relates the flexural rigidity of the plate (that resists bending) to the reaction of the fluid it rests upon (that sustains bending). The latter gives a measure of the distance over which the curvature of post-fracture fragments is different from the curvature of the original floe that gave rise to these fragments. We show an example of this in Fig. 3b.

We introduced the symbols $\kappa^<(x)$ and $\kappa^>(x)$ to denote the curvature of the left and right post-breakup fragments, with $x \in [0, L]$. By definition, $\kappa^<$ (respectively $\kappa^>$) exists only for $x \in [0, x_{fr}]$ (respectively $x \in [x_{fr}, L]$). We choose this integral definition of $L_\kappa$ because the differences in pre- and post-breakup curvatures is well-represented (when moving away from the fracture location) by a damped sine with oscillation period and attenuation rate $\sqrt{2}L_D$, that is,

$$\kappa(x) - \kappa^>(x) \sim \sin\left(\frac{x}{\sqrt{2}L_D}\right)\exp\left(-\frac{x}{\sqrt{2}L_D}\right),$$
(28)

which ensues from the shape of the solution presented in Appendix A1. Therefore, except at the floe boundaries where curvature is $0\,\mathrm{m}^{-1}$ (as imposed by the boundary condition, Eq. (6b)), the whole length of the initial floe may participate in releasing

energy. For long enough waves, the relaxation length tends to $\sqrt{2}L_D$, that is $\lim_{k \to 0} L_\kappa = \sqrt{2}L_D$. This can be shown analytically by assuming Eq. 28.

We thus have three typical horizontal length scales:

- The wavelength $\lambda = \frac{2\pi}{k}$, imposed by the wave forcing, and linearly tied to the domain length $L$ through the harmonic number $n$, so that $L = \frac{n\lambda}{2}$.

- The flexural length $L_D$, that depends on the properties of the material, the density of the fluid, and gravity. Only the former are varied in this study, with stiffer, thicker materials having a longer $L_D$.

- The relaxation length $L_\kappa$, that quantifies the distance over which fracture modifies the system.

As we consider short wavelengths and a very thin plate, capillarity effects could in principle be important. However, because the flexural length of the material exceeds its capillary length, these are negligible, which Auvity et al. (2025) verified experimentally. Therefore, we will not consider them either, and the dispersion relation we will be using is given in Equation (19), dominated by the term in $L_D{}^4$.

### 3.2 Linearity limitation

The analogue material used in the laboratory experiments of Auvity et al. (2025) requires nonlinear waves ($ak \approx 0.14$) for fracture to occur. As neither nonlinear plate nor non-linear waves are represented by our numerical model, we have to relax this condition, typically quantified by the wave slope $ak$, to observe fracture at all. Thus, we set the upper bound of our dichotomic searches so that $ak \leq 0.5$, which places us out of the linear framework our model relies on. As here, we are qualitatively showcasing the behaviour of our model rather than quantitatively exploiting the results, we deem this limitation to be inconsequential. For thickness and Young's modulus typical of sea ice, fracture in our model does happen in a linear regime, as illustrated in Fig. 3, where $ak = 0.015$.

Note that we define wave slope with respect to the wave propagating underneath the elastic plate and not with respect to the free surface waves. For a given time period, hydroelastic waves with dispersion relation Eq. (19) are typically longer than free surface gravity waves with dispersion relation Eq. (18), making the former slightly less steep.

### 4 Results

The results presented in this section focus on detecting a fracture threshold using our energy formalism and comparing this threshold to that obtained in the laboratory experiment of Auvity et al. (2025). To do so, we use in our simulations the material parameters issued from Auvity et al. (2025). Those are given in Table 1. We do not tune model parameters. As we are interested in detecting the fracture threshold, and our model does not have a fatigue term, we work with strictly unrelated quasi-static states, and we find the critical amplitude $a_{cr}$ systematically by dichotomic search. In Sect. 4.1, we detail the influence of varying the wavenumber exclusively. Then, in Sect. 4.2, we replicate the same protocol, while also varying the mechanical parameters of the plate.

**Table 1.** List of model parameters and their values. Parameters followed by an asterisk are inferred from other fixed parameters.

| Parameter | Value |
|---|---|
| density (fluid) | $1000 \, \mathrm{kg \, m^{-3}}$ |
| density (plate) | $680 \, \mathrm{kg \, m^{-3}}$ |
| energy release rate | $174 \, \mathrm{mJ \, m^{-2}}$ |
| flexural length* | $7.50 \, \mathrm{mm}$ |
| flexural rigidity | $3.1 \times 10^{-5} \, \mathrm{Pa \, m^3}$ |
| harmonic number | $3$ |
| Poisson's ratio | $0.4$ |
| thickness | $158 \, \mathrm{\mu m}$ |
| wavenumbers | $21.0$ to $203 \, \mathrm{rad \, m^{-1}}$ |
| Young's modulus* | $79.2 \, \mathrm{MPa}$ |

## 4.1 Reference case

We start by illustrating the response of our model, for a range of prescribed wavenumbers, with four quantities: the normalised position of the fracture $\frac{x_{\mathrm{fr}}}{L}$, the critical amplitude and curvature, and the relaxation length. These are presented in Fig. 6.
To exemplify the deviation between our computed deflection field and the forcing fluid surface, we choose here the case of the third harmonic, so that $L = \frac{3\pi}{k}$. We thus obtain curvature profiles that are symmetric[2] with respect to $x = \frac{L}{2}$, with three antinodes, which can be seen in Fig. 7. From left to right, the first and third antinodes (close to the left and right edges of the plate, respectively) are more influenced by the boundary conditions than the second one (located at the middle of the plate).

In what follows, we multiply $k$ by the flexural length, so that to obtain the (dimensionless) wavenumber $kL_D$. This allows exploring the model behaviour between two limits: small $kL_D$ values thus correspond to longer waves and lower flexural rigidity (the plate conforms to the fluid), while high values correspond to shorter waves and higher flexural rigidity (the plate is non-deformable). We divide Fig. 6b–d in four regions, based on different behaviours of the fracture location as predicted by the energy criterion, identified in Fig. 6a. These regions are separated by $kL_D = 0.1638$, $0.3275$ and $0.7578$. They correspond, for increasing $kL_D$, to fracture happening in the middle of the floe (the second curvature antinode); fracture happening close to the first or third curvature antinode; fracture again happening in the middle of the floe; and fracture uncorrelated from any curvature extremum. Because of the particular wave forcing imposed in our model configuration, the free energy profile is symmetrical with respect to the middle of the plate. Therefore, it is not numerically possible to discriminate between an energy minimum happening at $x_{\mathrm{fr}}$ or $L - x_{\mathrm{fr}}$, and in Fig. 6a, we only show the branch corresponding to $x_{\mathrm{fr}} \leq \frac{L}{2}$.

We note that, in regions 1 to 3, fracture predicted by the energy criterion does not systematically happen at the global curvature extrema. In this third harmonic case, the global extremum is in the middle of the floe, except in the band $kL_D \in$

---

[2]As would be the case for any odd harmonic number. In the case of even harmonic number, the curvature profile as a twofold rotational symmetry about $(x, \kappa) = \left(\frac{L}{2}, 0\right)$. In other words, $\kappa\left(x - \frac{L}{2}\right)$ is even-symmetric for odd harmonic wavenumbers, and odd-symmetric for even harmonic numbers.

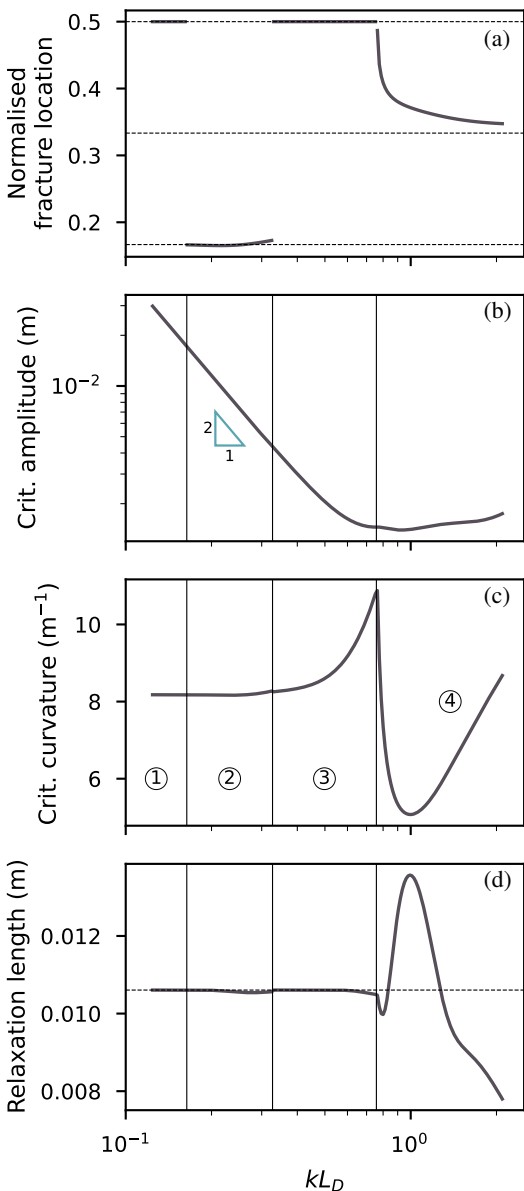

**Figure 6.** Relationships between the nondimensionalised wavenumber and normalised (with respect to plate length) fracture location (a), critical amplitude (b), critical curvature (c), and relaxation length (d). Model parameters are provided in Table 1. In (a), three horizontal dashed lines represent the asymptotes $\frac{x_{\text{fr}}}{L} = \frac{1}{6}$, $\frac{1}{3}$ and $\frac{1}{2}$. In (b) to (d), vertical lines show delimitations between regions corresponding to different behaviours of fracture location, observed on (a). The regions are numbered in (c). In the second region of (b), the triangle of height twice its horizontal base (in loglog space and data units) gives an indication of the slope. In (d), an horizontal dashed line represents the asymptote $\sqrt{2}L_D$.

[0.178, 0.357]. The overlap with the region 2, defined by $kL_D \in [0.1638, 0.3275]$, is thus not one-to-one. Additionally, in region 2, fracture does not happen at the antinode, but in its vicinity. For $kL_D < 0.242$, fracture is on the left of the first antinode, and for $kL_D > 0.243$, to its right (the situation is reversed for the third antinode). In region 4, fracture happens far from either antinode. It seems that for increasing $kL_D$, $\frac{x_{\mathrm{fr}}}{L} \to \frac{1}{3}$, which corresponds to a node of the forcing.

In Fig. 7, we show examples of the behaviour of the free energy and of the along-plate curvature for these different regions. We compare the latter to the "conforming" curvature, $\kappa_{\mathrm{conf}}(x) = \frac{\mathrm{d}^2}{\mathrm{d}x^2} w_{\mathrm{conf}}$, with $w_{\mathrm{conf}}(x) = \frac{a}{1+(kL_D)^4} \sin(kx)$ the associated displacement stemming from the fluid surface. The term at the denominator ensures that it satisfies Eq. (6a), and for long waves, $\lim_{k \to 0} w_{\mathrm{conf}} = \eta$. In Fig. 8, we show for the same examples the floe deflection, compared to the forcing amplitude, and we indicate the relaxation length. To aid comparison, we normalise the along-floe coordinate by the floe length.

In the first two regions, both corresponding to small wavenumbers, the difference between curvature and conforming curvature (Fig. 7a, Fig. 7b) is noticeable only in the immediate vicinity of the edges of the domain. Minima of free energy correspond roughly with extrema of curvature, and the floe deflection (Fig. 8a, Fig. 8b) follows the fluid surface. In the first region, and to a lesser degree in the second region, the critical curvature (Fig. 6c) varies little, although a slight positive trend exists. Our strain-based and energy-based criteria in these two regions would therefore predict virtually similar fractures. The critical am-

plitude (Fig. 6b) varies with the inverse of the squared wavenumber, that is, with the square of the wavelength. The relaxation length (Fig. 6d) is almost constant and tends to $\sqrt{2}L_D$ from below for decreasing wavenumbers. As the floe length is, in the case of stationary wave forcing, inversely proportional to the wavenumber, the relaxation length *normalised* by the floe length increases with the wavenumber, and is not constant across the different panels of Fig. 8.

     In the third region, curvature and conforming curvature (Fig. 7c) are now dissimilar between the left (respectively right) edge and the left (respectively right) antinode. The free energy still shows three troughs, but the trough at $x = \frac{L}{2}$ is now clearly more pronounced than the other two. There are still three distinct deflection extrema (Fig. 8c), synchronised with the forcing wave, and the amplitude of deflection of the floe is slightly smaller than the forcing amplitude. From $kL_D \approx 0.3057$, we locally (around the two positive deflection antinodes) have $\eta - w > h - d$. As $h - d$ corresponds to the freeboard of the floe at rest, this suggests parts of the deformed floe are immersed. This takes place close to the transition from the second region,

which happens at $kL_D = 0.3275$. The non-zero deflection near the edges shows that deflection is now significantly different from the sine forcing. In terms of the occurrence of fracture, this region corresponds to a sharp increase in critical curvature, incompatible with a strain-based (that is, constant critical curvature) criteria. The relaxation length, however, is still practically constant with $kL_D$, and a good indicator of the zone over which pre- and post-fracture modelled deflection differ. An inflexion of the critical amplitude decrease rate is also visible. In regions 1 to 3, fracture locations near antinodes and the shape of

post-fracture deflections are consistent with mode I fracturing.

     As $kL_D$ increases, the length of the plate diminishes relatively to its flexural length. The impact of the boundary conditions on the deflection profile is therefore amplified. This effect is sizeable in the fourth region (Fig. 8d): curvature and conforming curvature (Fig. 7d) no longer match anywhere along the plate. This is despite staying in a regime where $L_D \gg L$, as $kL_D = n\pi \frac{L_D}{L} \approx 10 \frac{L_D}{L}$. The central free energy trough has separated into two distinct troughs corresponding to global minima, no

longer in phase with curvature extrema. This separation corresponds to the transition from the third region. The fourth region

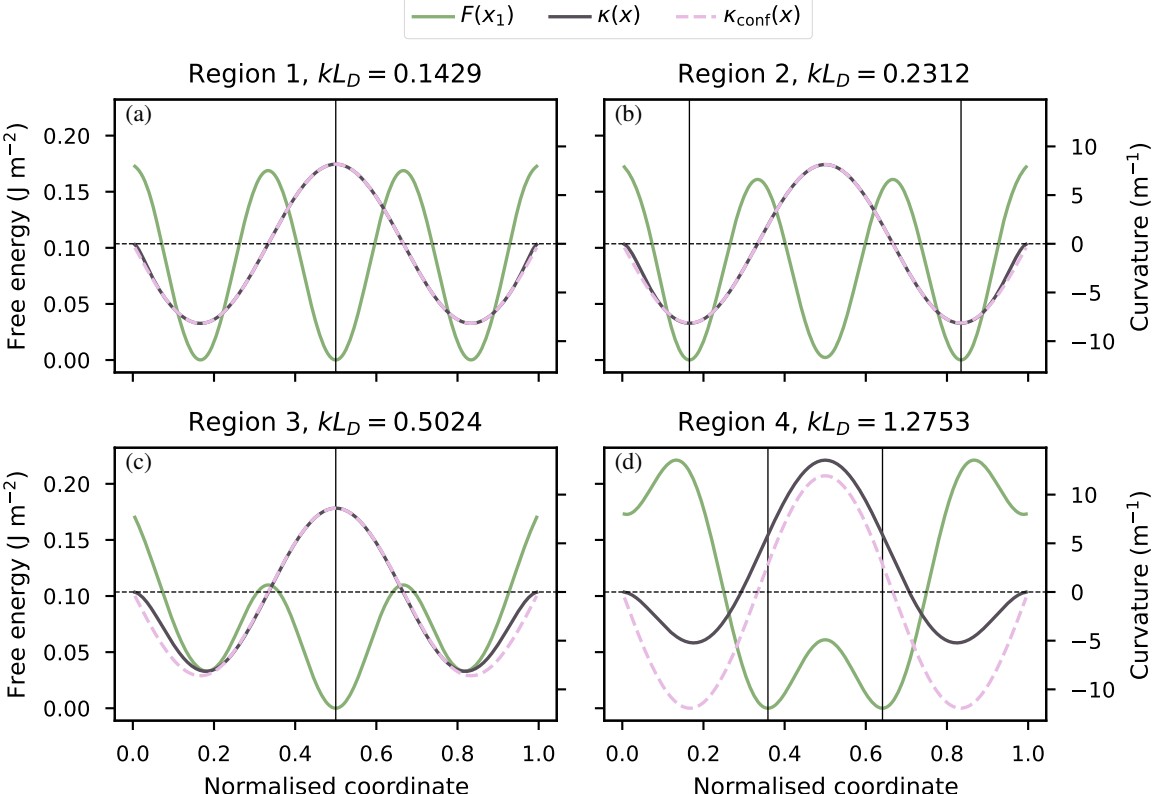

**Figure 7.** Examples of free energy (left vertical axes, green lines) and curvature (right vertical axes, grey lines) profiles for the four regions identified in the text. A sine curvature is shown in dashed lines as a comparison to the curvature derived from Eq. (8). The fracture locations, determined from the energy criterion, are shown with vertical lines: the energy profiles being symmetrical with respect to the middle of the plates, $F(x_{\mathrm{fr}}) = F(L - x_{\mathrm{fr}})$. The dashed horizontal lines show the zero-curvature reference.

also shows a drop in critical curvature (Fig. 6c), which has been monotonically increasing with $kL_D$ thus far. However, the maximum curvature $\kappa(L/2)$ keeps increasing irregularly with $kL_D$, as can be seen by comparing Fig. 8c and Fig. 8d. The critical amplitude, which seems to plateau on the right of the third region (Fig. 6b), is singular at the transition, then diminishes again before increasing irregularly. Additionally, the edges of the fragments no longer mirror each other. There is a significant post-fracture discontinuity in deflection, which is a characteristic of region 4, and not consistent with bending (mode I) fracture, but reminiscent of a sliding (mode II) or tearing (mode III) fracture. A minimum of critical curvature is reached for $kL_D = 1$, which also corresponds to a maximum of relaxation length. For $n \geq 2$, there is a single positive $kL_D$, quickly converging to 1, for which the slope at the edges of the floe vanishes, that is, $\left(\frac{\mathrm{d}w}{\mathrm{d}x}\right)_{x=0,L} = 0$. It corresponds to the two outside-most deflection antinodes vanishing, leaving only the internal ones. It can be seen from the deflection profile in Fig. 8d, that compared to Fig. 8a–c, the slope at the edges has changed sign, and that a single antinode (at the centre of the floe) remains.

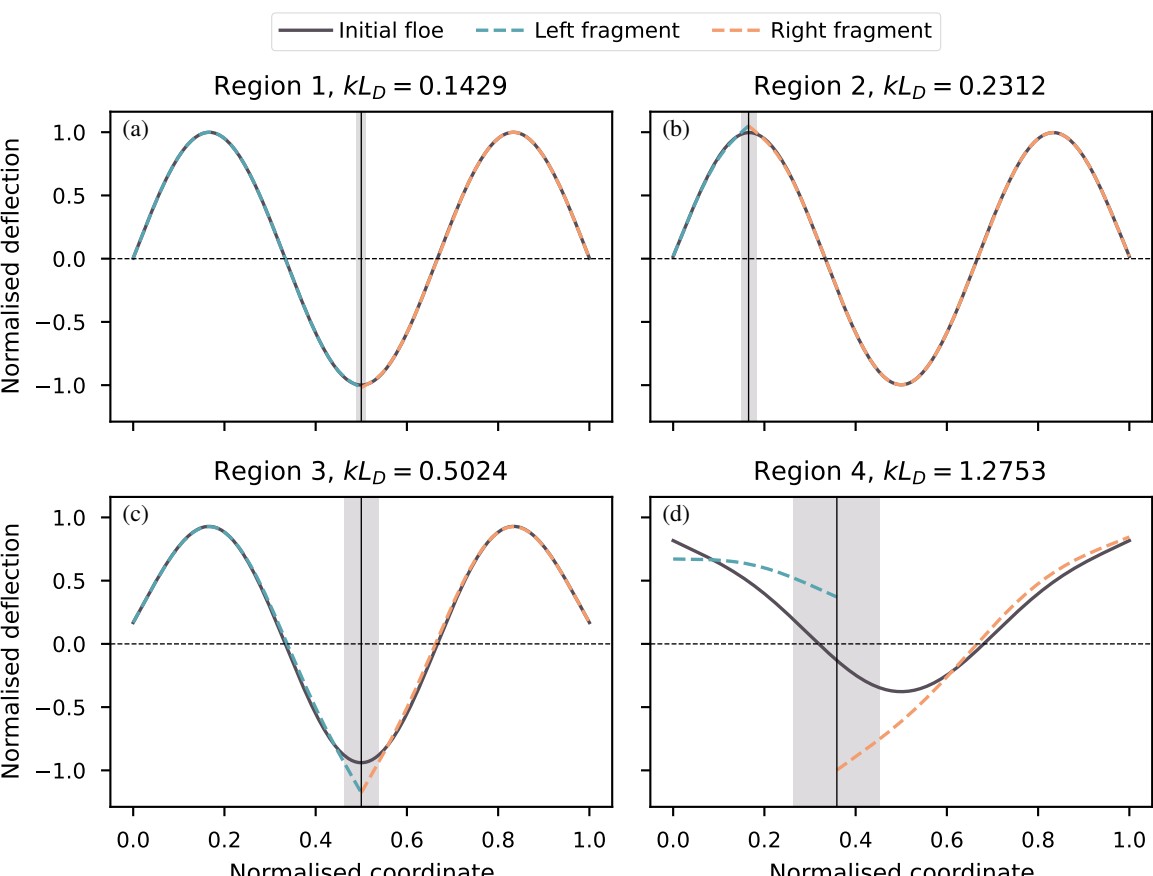

**Figure 8.** Examples of pre- and post-fracture floe deflection profiles for the four regions identified in the text, normalised by the critical amplitude. The relaxation lengths, centred on the fracture locations, are shown with shaded rectangles. We only consider the left fracture location, where applicable. The dashed horizontal lines show the zero-deflection reference.

## 4.2 Influence of mechanical parameters

We further investigate the response of our model to varying mechanical parameters, by reproducing the analysis presented in Sect. 4.1 for an ensemble of $(h, Y)$ pairs with 128 members. Doing so, we aim to reproduce the internal variability that stems from laboratory conditions in the experiments of Auvity et al. (2025). We generate this ensemble through Latin hypercube sampling, and enforce that the two variables are independent with normal marginal densities of prescribed means $100\,\mu\mathrm{m}$ and $70\,\mathrm{MPa}$, and prescribed standard deviations $20\,\mu\mathrm{m}$ and $14\,\mathrm{MPa}$, respectively; we show the joint density of our sample in Fig. 9. The resulting distribution of flexural rigidities is positively skewed, with mean $7.88 \times 10^{-6}\,\mathrm{Pa\,m^3}$ and median $6.93 \times 10^{-6}\,\mathrm{Pa\,m^3}$.

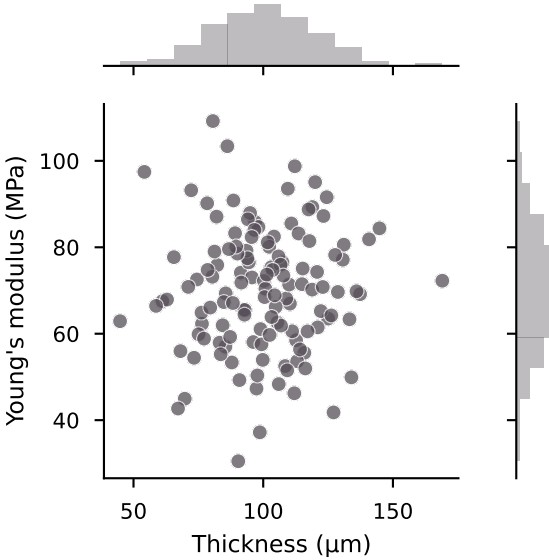

**Figure 9.** Joint density and marginal densities of our $(h, Y)$ ensemble.

We further impose the relation

$$\frac{2Gh^2}{D} = C \tag{29}$$

between the energy release rate, the Young's modulus, and the thickness, as derived by Auvity et al. (2025), setting the dimensionless material constant parameter $C = 2.8 \times 10^{-4}$. This expression serves as a proxy establishing a value for $G$, which is poorly constrained experimentally. The other parameters are kept fixed at the values presented in Table 1. We also keep the same constraint on the wave slope, requiring $a_{\mathrm{cr}}k \le 0.5$: depending on the precise values assumed by the thickness and Young's modulus, the interval of wavenumbers that leads to fracture may vary.

### 4.2.1 Comparison to experimental data

We show numerical results in Fig. 10 (colour-coded lines), which we compare to experimental data from Auvity et al. (2025) (circle and square markers). We obtain results similar to those presented in Sect. 4.1. The critical amplitude profiles do not depend on the mechanical properties of the simulated material, up to a multiplicative constant, that increases with flexural rigidity (or, equivalently, flexural length).

This fact extends to the other variables shown in Fig. 6. Within the ranges of mechanical parameters explored, which are in agreement with the values and internal variability estimated by Auvity et al. (2025), the order of magnitude of the critical amplitude and its decreasing tendency with increasing $kL_D$ agree between the simulations and the laboratory experiments. However, and as expected, this agreement is only qualitative. Indeed, we recall that in the experimental setting, fracture was only obtained with nonlinear waves. It is likely the reason why the critical amplitudes measured by Auvity et al. (2025) varies as

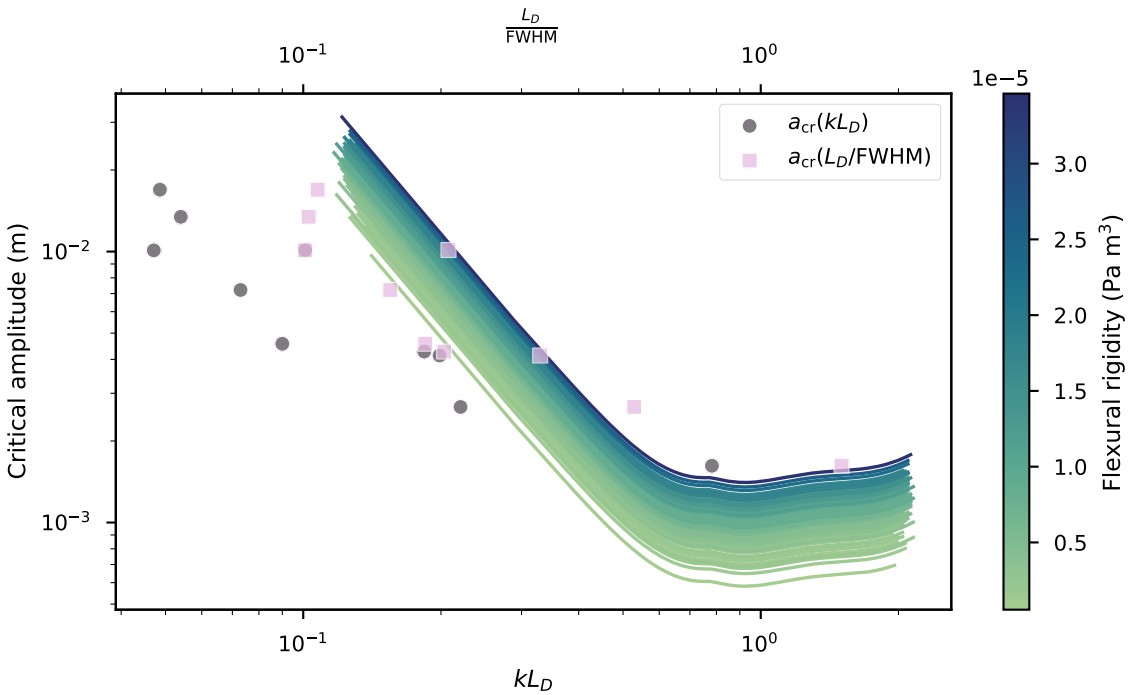

**Figure 10.** Relation between the dimensionless wavenumber and the critical amplitude, for varying thickness and Young's modulus. Lines are numerical results, with the colour scale indicating the flexural rigidity, that combines the two varying parameters. Grey dots are experimental results from Auvity et al. (2025). Pink squares are the same experimental results, with a different horizontal scaling, as described in the text.

$k^{-1}$, not as $k^{-2}$ as we find numerically. The differences between experimental and numerical critical amplitudes are therefore deepened at small wavenumbers.

We also note that the way we define the relaxation length $L_\kappa$ in our linear waves simulations differs from the definition of Auvity et al., who used the full width at half-maximum (hereinafter, FWHM) of pre-fracture curvature in their nonlinear
experiments. On Fig. 10, we thus also represent the experimental critical amplitudes as a function of $\frac{L_D}{\text{FWHM}}$ (pink squares), which horizontally shifts the experimental points, in an attempt to correct the discrepancy between their nonlinear waves forcing and our linear model.

We do not adopt their definition, as it would be incompatible with our region 4 results, where fracture does not happen around curvature peaks or troughs, while experimental fractures always happened in the vicinity of a deflection antinode. If we
were to apply it to regions 1 to 3, we would obtain something very similar to the FWHM of $\sin(kx)$, that is $\frac{2\pi}{3k}$, $\frac{\lambda}{3}$. Because of their nonlinear wave forcing, Auvity et al. measured FWHMs that varied like $\frac{\lambda}{12}$. Bending was thus concentrated in a smaller fraction of their forcing wavelengths, and $\frac{L_D}{\text{FWHM}}$ can be seen as an alternative, rescaled dimensionless wavenumber. Using this definition, we can improve the overlap between experimental and numerical results, with experimental points falling in

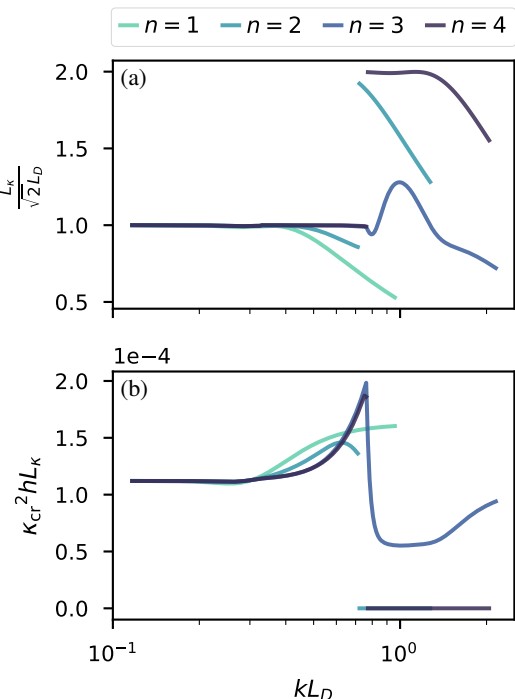

**Figure 11.** Relationship between the nondimensionalised wavenumber and two dimensionless quantities: the energy dissipation length scaled by the flexural length (a), and squared critical curvature scaled by thickness and energy dissipation length (b). We show ensemble averages, for four harmonic numbers. For a given harmonic number, ensemble members are virtually identical, with coefficients of variation well below $1 \times 10^{-3}$ where the means are non-zero.

regions 1 to 3, where both model end experiments show the critical curvature depends on the forcing wavelength, precluding a
constant strain threshold.

### 4.2.2 Impact of harmonic number and dimensionless quantities

So far, we focused on the harmonic number $n = 3$. However, for large enough $kL_D$, the response of the model depends on $n$, and therefore on the geometry of the domain. In particular, for even $n$, fracture in region 4 happens systematically in the middle of the floe. Due to the symmetry property of the forcing, this means our model predicts fracture with $\kappa_{cr} = 0\,\mathrm{m}^{-1}$,
inconsistent with bending fracture but reminiscent of shearing or tearing fracture. The fundamental configuration, with $n = 1$, is a particular case. The maxima of deflection, curvature, and free energy happen at $x = \frac{L}{2}$ independently of $kL_D$. We illustrate this difference in Fig. 11 by presenting the relationships between the nondimensionalised wavenumber and two dimensionless quantities, for different harmonic wavenumbers. We choose these two quantities because they exhibit the remarkable property of depending on the dimensionless wavenumber, but not on the individual variations of the mechanical parameters.

The first of these quantities, shown in Fig. 11a, is the relaxation length $L_\kappa$ (as shown in Fig. 6d for the fixed parameters experiment), normalised by the augmented flexural length $\sqrt{2}L_D$. We retrieve, independently of the harmonic number, the limit $\lim_{k\to 0} L_\kappa = \sqrt{2}L_D$ already stated in Sect. 3.1. Behaviours depending on the harmonic number emerge from $kL_D \approx 0.4$. If all curves show a downward trend in what corresponds to $kL_D$ in regions 2 and 3, this trend is more pronounced for smaller numbers, in particular $n \in \{1, 2\}$. The second striking difference, is that between even and odd harmonic numbers. There is a discontinuity at the transition from region 3 to region 4 for even numbers, with an upward jump preceding a sustained downward trend. This transition is continuous for $n = 3$. There is no region 4 behaviour for $n = 1$, as in this configuration, fracture always happens at $\frac{x_{\mathrm{fr}}}{L} = \frac{1}{2}$.

The second dimensionless number, shown in Fig. 11b, can be built as the product of two distinct dimensionless quantities: critical curvature multiplied by thickness (that is, twice the critical strain), and critical curvature multiplied by relaxation length. Notably, both these quantities do depend on thickness and Young's modulus, without showing the ordering critical amplitude does in Fig. 10. However, their product, $\kappa_{\mathrm{cr}}{}^2 h L_\kappa$, only depends on $kL_D$. This quantity was also derived by Auvity et al. (2025), who interpreted it as a constant independent of the wave forcing. We do not replicate this result outside of regions 1 and 2, that is, the wavenumber band where neither critical curvature nor relaxation length vary. As in Fig. 11a, the different curves are indistinguishable within these two regions (that is, at small $kL_D$), and a discontinuity exists for even harmonic numbers between regions 3 and 4, as for these, the critical curvature drops to $0\,\mathrm{m}^{-1}$ in region 4.

Finally, it can be seen that the upper bound of the range of $kL_D$ that sees fracture happens depends on $n$. It first increases with $n$ but peaks for $n = 3$, and then decreases. This is despite keeping the same $a_{\mathrm{cr}}k \le 0.5$ criterion on the wave slope. The lower bound, however, does not change and keeps the value $kL_D = 0.1167$. The differences between the different harmonics in region 4 are explained by the loss of deflection extrema near the boundaries as $kL_D$ increases, which have dissimilar effects on the deflection profile in this region for different $n$. For $n > 2$, there exists a single $kL_D$ for which the slope of the deflection at the edges of the plate, $\left(\frac{\mathrm{d}w}{\mathrm{d}x}\right)_{x=\{0,L\}}$, cancels. It converges exponentially towards 1, so that noting it $(kL_D)_0$, we have $\log|1 - (kL_D)_0| \sim -n$. The cancellation of the slope conveys the transition from a deflection profile with $n$ antinodes to one with $n - 2$ antinodes. When $kL_D$ keeps increasing, the higher $n$, the higher the variety of behaviours shown by $w$. However, for large enough $kL_D$, $w(x) \to \frac{2a}{n\pi}$ for odd $n$, and $w(x) \to \frac{2a}{n}\left(-\frac{2x}{L} + 1\right)$ for even $n$. These are rigid motions, independent of the forcing, corresponding respectively to heave (translation) or pitch (rotation).

## 5   Discussion and conclusion

We have developed a versatile, lightweight one-dimensional model that simulates the time-dependent fracture of sea ice by waves. This model has the particularity of solving directly for the deflection of a floe caused by the competition between buoyancy and gravity; instead of solving for waves scattered by the presence of the floe at the ice–fluid interface, and assuming that the deflection follows that interface. We can thus use the model to continuously explore the response of a floe to bending between two limits: an elastic plate conforming exactly to a fluid foundation, and an undeformable plate. We have implemented in this model two fracture criteria. One, compatible with continuum fracture mechanics, is based on looking for a global post-

fracture energy minimum and comparing it to the pre-fracture energy state to determine whether fracture should occur, and is a novelty of this model. The other is compatible with the more common hydroelastic approach applied to sea ice, based on locally comparing strain to a prescribed, constant threshold.

The present study is centred on presenting the theoretical and numerical aspects of the model itself and validate/invalidate the energy-based fracture criterion. We apply the model to an analogue material used in the laboratory to study wave-induced ice fracture, in the specific setting of monochromatic stationary waves. Because no constant critical strain threshold was observed during these experiments, but a relationship between energy release rate and other parameters of our model exists, we focus on investigating the energy criteria. Even in this particularly simplified configuration, the response of the model in terms of critical amplitude or curvature is not straightforward.

Our results indicate that the critical curvature derived from an energy-based fracture depends on the forcing wave, contradicting the existence of a universal critical strain. This was also observed in the laboratory (Auvity et al., 2025). In the (dimensionless) wavenumber band where our results overlap with the experimental data from Auvity et al. (2025), we obtain comparable critical amplitudes. However, we are not able to replicate their scaling for low $kL_D$. Additionally, we obtain that for large enough $kL_D$ (region 4), the two criteria, energy-based and critical strain-based, diverge on the predicted fracture location, in that energy-predicted fracture is uncorrelated from curvature, or strain, extrema. The fracturing behaviour in region 4 is inconsistent with bending fracture, and suggests out-of-plane shear or in-plane shear fracturing. The latter would describe fracture propagating perpendicularly to the direction of wave propagation (that is, as someone tearing up a sheet of paper), in contradiction with the invariance hypothesis made on the modelled plate and therefore cannot be represented in the current 1D model.

A possible explanation for the different relationship between critical amplitude and wavenumber observed experimentally ($a_{\mathrm{cr}} \sim k^{-1}$) and numerically ($a_{\mathrm{cr}} \sim k^{-2}$, for $kL_D \lesssim 0.6$) may be that experimentally, for fracture to happen, the material considered (varnish) required nonlinear wave forcing, which our model does not represent. In the case of ice, the linearity assumption is, however, valid. For values corresponding to a similar experiment conducted on fresh water ice by the same team (personal communication with Auvity et al.), that is thicker and stiffer than their varnish, our results in terms of critical amplitude as a function of dimensionless wavenumber are, up to multiplicative constants, identical to those presented here, and the wave slopes required to obtain fracture are typically of the order of $0.02$; well within the linear regime. Therefore, increasing the numerical complexity of the model to accommodate nonlinear plate behaviour seems unnecessary at this stage. Another explanation is that, while our model considers homogeneous plates with constant Young's modulus, the material engineered by Auvity et al. (2025) is obtained by layering. Because of introduced vertical inhomogeneities, this process is likely to introduce a dependency of the Young's modulus to the obtained thickness. Further analysis of this new dataset, and in particular whether the dependency of the curvature at failure on the wavenumber exists, is ongoing.

The key features of our model are that bending is driven exclusively by the along-plate variation of buoyancy, which cannot be resolved by hydroelastic models, and that fracture can be controlled by an energy criterion integrated over the entire floe. For high $kL_D$ values, that is either large wavenumber or a stiff elastic plate, this leads to physically questionable behaviours, such as submergence, and post-breakup deflection discontinuity across the fracture (region 4) or even fracture at zero-curvature

(for even harmonics). We note that the possibility of submergence is the direct consequence of the weak, one-way coupling between fluid and plate, as we only represent the response of the plate to the fluid, while ignoring the feedback response of the fluid. This one-way coupling is a trade-off allowing us to maintain the theoretical and numerical complexity low. As a rationality check, we verify that the bending energy, transmitted to the plate by the fluid, is orders of magnitude less than the gravitational potential energy of the fluid: there is thus no unaccounted energy leaks into the plate. As none of this energy is returned to the fluid, it is likely we overestimate the likeliness of fracture.

Here, we have described the simulated behaviour as a function of $kL_D$ by distinguishing the results in 4 different regions, based on where the fracture happens. The $kL_D$ thresholds being regions should however be regarded carefully, as they depends slightly on the harmonic number of the forcing, and might be a feature of stationary forcing. At field scale, with waves propagating within the ice cover, the fracture front follows the wave front, in such a way that fragments are typically smaller than the dominant wavelength (Dumas-Lefebvre and Dumont, 2023). Therefore, the increased complexity in the model behaviour at high harmonic numbers may not be representative of natural conditions.

More experimental data is needed to confirm or infirm the behaviour of the model in the large $kL_D$ band, and whether a forcing-dependent trend for critical curvature exists in the case of ice. Previous wave tank fracture experiments (Dolatshah et al., 2018) showed bending failure typically happening at $kL_D \approx 1$, though the uncertainties on thickness and Young's modulus are quite large. Other values of Young's modulus reported for such experiments, in the low $\mathrm{MPa}$ range (Herman, 2018; Passerotti et al., 2022), seem inconsistent with a cohesive, solid sheet of ice, as represented in our model. Nevertheless, these authors did observe fracture in the range $kL_D \in [0.29, 0.54]$ and for $kL_D = 0.29$, respectively. Voermans et al. (2020) compiled a list of studies of wave-induced breakup observations. Mechanical parameters were, for the most part, not measured, but they suggest estimations based on known empirical relations. Following their methods, we can generate ensembles of $kL_D$ wavenumbers, that we find lying in the range $kL_D \in [1.7 \times 10^{-2}, 3.1]$ for both breaking and non-breaking cases. In the case of realistic wave forcing, with material parameters representative of first-year sea ice, the peak of wave energy occurs in $kL_D$ bands corresponding to what we identified as region 3 and 4. However, higher-frequency waves are also the ones most effectively attenuated by the ice cover, so that they contribute less to fracture.

We acknowledge our results are a first step towards the validation of the fracture formalism we propose. Planned future work will involve using our model to study whether the choice of fracturing criterion impacts the floe size distribution resulting from propagating wave-induced breakup, and applying it in configurations corresponding to recent and exciting observations of transient wave-induced breakup of instrumented ice in a natural setting (Kuchly et al., 2025).

*Code and data availability.* The current version of SWIIFT is available from the project website https://github.com/sasip-climate/swiift under the APACHE-2.0 licence. The exact version of the model used to produce the results used in this paper is archived on https://doi.org/10.5281/zenodo.15528673 (Mokus, 2025d). The input data and scripts to run the model and produce the plots for all the simulations presented in this paper are archived on https://doi.org/10.5281/zenodo.15528650 (Mokus, 2025c). A package dedicated to reproducing the figures, holding the necessary data but no model logic, is archived on https://doi.org/10.5281/zenodo.15230102 (Mokus, 2025b).

*Video supplement.* An animation of the simulation of fracture by a spectral wave forcing, as described in the text, is available at Mokus (2025a).

## Appendix A:  Equation for the moment-deformation

We consider the boundary problem

$$
\begin{cases}
\dfrac{\mathrm{d}^4 w}{\mathrm{d}x^4} = \sigma^4 \big[\eta(x) - w(x)\big] & x \in [0, L] & \text{(A1a)} \\[2ex]
\dfrac{\mathrm{d}^2 w}{\mathrm{d}x^2} = 0 & x \in \{0, L\} & \text{(A1b)} \\[2ex]
\dfrac{\mathrm{d}^3 w}{\mathrm{d}x^3} = 0 & x \in \{0, L\}. & \text{(A1c)}
\end{cases}
$$

where $\eta$ denotes the surface undergoing wave forcing, $w$ the floe deflection, and $\sigma = \frac{1}{L_D}$ is the reciprocate of the flexural length.

The surface $\eta$ is typically the superposition of propagating, attenuated wave modes, so that

$$
\eta(x) = \sum_j \eta_j(x) \tag{A2}
$$

and

$$
\eta_j(x) = a_j \exp(-\alpha_j x) \sin(k_j x + \phi_j), \tag{A3}
$$

with $a$ wave amplitude, $\alpha$ wave attenuation per unit distance, $k$ wave number, and $\phi$ wave phase at the left floe edge. The index $j$ is used with respect to a discretised wave spectrum.

Finally, we define the elastic energy per unit cross-sectional area

$$
E = \frac{D}{2h} \int_0^L \left( \frac{\mathrm{d}^2 w}{\mathrm{d}x^2} \right)^2 \mathrm{d}x. \tag{A4}
$$

In the rest of this document, we will note

$$
\kappa(x) = \frac{\mathrm{d}^2 w}{\mathrm{d}x^2} \tag{A5}
$$

the curvature of the floe.

The ODE (A1a) is linear, and so are the boundary conditions fourth order ODE, but it is linear, and so are the boundary conditions (A1b),(A1c). Here, we consider the simplified case where $D$ and $h$ are constant, making Eq. (A1a) a constant-coefficients, linear ODE.

### A1  Solution to the BVP on the floe deflection

The ODE (A1a) is linear and non-homogeneous. Its general solution $w$ is the superposition of an homogeneous solution $w_h$ and a particular solution $w_p$.

 **A1.1   Homogeneous solution**

The characteristic polynomial of the homogeneous ODE

$$\frac{\mathrm{d}^4 w}{\mathrm{d}x^4} + \sigma^4 w(x) = 0 \tag{A6}$$

associated with (A1a) is

$$P(s) = s^4 + \sigma^4. \tag{A7}$$

It has solutions

$$(1+\mathrm{i})\tilde{\sigma}, (1-\mathrm{i})\tilde{\sigma}, (-1+\mathrm{i})\tilde{\sigma}, (-1-\mathrm{i})\tilde{\sigma} \tag{A8}$$

with $\tilde{\sigma} = \frac{\sqrt{2}}{2}\sigma$. Independent solutions to (A1a) are thus

$$\check{\boldsymbol{f}} = \begin{bmatrix} \exp(\tilde{\sigma}x)(\cos(\tilde{\sigma}x) + \mathrm{i}\sin(\tilde{\sigma}x)) \\ \exp(\tilde{\sigma}x)(\cos(\tilde{\sigma}x) - \mathrm{i}\sin(\tilde{\sigma}x)) \\ \exp(-\tilde{\sigma}x)(\cos(\tilde{\sigma}x) + \mathrm{i}\sin(\tilde{\sigma}x)) \\ \exp(-\tilde{\sigma}x)(\cos(\tilde{\sigma}x) - \mathrm{i}\sin(\tilde{\sigma}x)) \end{bmatrix}. \tag{A9}$$

Applying the full-rank linear transformation

$$675 \quad \frac{1}{2}\begin{bmatrix} 1 & 1 & 0 & 0 \\ -\mathrm{i} & \mathrm{i} & 0 & 0 \\ 0 & 0 & 1 & 1 \\ 0 & 0 & -\mathrm{i} & \mathrm{i} \end{bmatrix} \tag{A10}$$

to $\check{\boldsymbol{f}}$ yields the real-valued independent solutions

$$\boldsymbol{f} = \begin{bmatrix} \exp(\tilde{\sigma}x)\cos(\tilde{\sigma}x) \\ \exp(\tilde{\sigma}x)\sin(\tilde{\sigma}x) \\ \exp(-\tilde{\sigma}x)\cos(\tilde{\sigma}x) \\ \exp(-\tilde{\sigma}x)\sin(\tilde{\sigma}x) \end{bmatrix}. \tag{A11}$$

Finally, any linear combination $\boldsymbol{f}^T \boldsymbol{c}$, with the real-valued vector

$$\boldsymbol{c} = \begin{bmatrix} c_1, c_2, c_3, c_4 \end{bmatrix} \tag{A12}$$

is a solution to (A6).

### A1.2 Particular solutions

The non-homogeneous term in (A1a) can be written

$$\sigma^4 \eta(x) = \sigma^4 \sum_j \eta_j(x) \tag{A13}$$

$$= \sigma^4 \sum_j \mathrm{Im}\left[\hat{a}_j e^{i\hat{k}_j x}\right] \tag{A14}$$

with the complex amplitudes $\hat{a} = a e^{i\phi}$ and the complex wavenumbers $\hat{k} = k + i\alpha$. Using the exponential response formula, and the characteristic polynomial (A7), we obtain particular solutions of the form

$$w_{p,j} = \sigma^4 \mathrm{Im}\left[\frac{\hat{a}_j e^{i\hat{k}_j x}}{\hat{k}_j^{\,4} + \sigma^4}\right] \tag{A15}$$

$$= \mathrm{Im}\left[\frac{\hat{a}_j e^{i\hat{k}_j x}}{\left(\hat{k}_j/\sigma\right)^4 + 1}\right] \tag{A16}$$

so that the particular solution to (A1a) can be written

$$w_p = \sum_j w_{p,j}. \tag{A17}$$

In what follows, we will write

$$\tilde{a}_j = \frac{\hat{a}_j}{\left(\hat{k}_j/\sigma\right)^4 + 1} \tag{A18}$$

the complex amplitude of these solutions. As $\sigma, k_j, \alpha_j > 0$, this amplitude exists only if

$$k_j \neq \tilde{\sigma} \wedge \alpha_j \neq \tilde{\sigma}. \tag{A19}$$

### A1.3 Coefficients of the homogeneous solution

The coefficients of $c$ have to be determined to enforce the boundary conditions. Enforcing these four conditions leads to the system

$$\mathbf{D}_{\mathrm{BC}}\mathbf{M}_{\mathrm{BC}}c = r \tag{A20}$$

with

$$\mathbf{D}_{\mathrm{BC}} = \begin{bmatrix} 2\tilde{\sigma}^2 & 0 & 0 & 0 \\ 0 & 2\tilde{\sigma}^2 & 0 & 0 \\ 0 & 0 & 2\tilde{\sigma}^3 & 0 \\ 0 & 0 & 0 & 2\sqrt{2}\tilde{\sigma}^3 \end{bmatrix}, \tag{A21}$$

$$
\mathbf{M}_{\mathrm{BC}} = \begin{bmatrix} 0 & 1 & 0 & -1 \\ -e^{\beta}\sin(\beta) & e^{\beta}\cos(\beta) & e^{-\beta}\sin(\beta) & -e^{-\beta}\cos(\beta) \\ -1 & 1 & 1 & 1 \\ -e^{\beta}\sin(\beta+\frac{\pi}{4}) & e^{\beta}\cos(\beta+\frac{\pi}{4}) & e^{-\beta}\cos(\beta+\frac{\pi}{4}) & e^{-\beta}\sin(\beta+\frac{\pi}{4}) \end{bmatrix},
\tag{A22}
$$

where $\beta = \tilde{\sigma}L$, and

$$
\boldsymbol{r} = \mathrm{Im}\sum_{j} \begin{bmatrix} \hat{k}_j^2 \tilde{a}_j \\ \hat{k}_j^{\,2}\tilde{a}_j e^{\mathrm{i}\hat{k}_j L} \\ \mathrm{i}\hat{k}_j^3 \tilde{a}_j \\ \mathrm{i}\hat{k}_j^3 \tilde{a}_j e^{\mathrm{i}\hat{k}_j L} \end{bmatrix}.
\tag{A23}
$$

The third and first lines of the system give

$$
\begin{cases} c_1 = \dfrac{1}{2\tilde{\sigma}^2}\left(r_1 - \dfrac{r_3}{\tilde{\sigma}}\right) + c_3 + 2c_4 & \text{(A24a)} \\[2ex] c_2 = \dfrac{r_1}{2\tilde{\sigma}^2} + c_4 & \text{(A24b)} \end{cases}
$$

and the system (A20) can be simplified to the more tractable

$$
\mathbf{M}_{\mathrm{II}} \begin{bmatrix} c_3 \\ c_4 \end{bmatrix} = \begin{bmatrix} -r_3 \\ r_4 \end{bmatrix}
\tag{A25}
$$

with

$$
\mathbf{M}_{\mathrm{II}} = 2 \begin{bmatrix} \sin(\beta)\sinh(\beta) & e^{\beta}\sin(\beta) - \cos(\beta)\sinh(\beta) \\ -\frac{\sqrt{2}}{2}\left[\sin(\beta)\cosh(\beta) + \cos(\beta)\sinh(\beta)\right] & -\frac{\sqrt{2}}{2}e^{\beta} - \sin(\beta+\frac{\pi}{4})\sinh(\beta) \end{bmatrix}.
\tag{A26}
$$

The determinant of $\mathbf{M}_{\mathrm{II}}$ is

$$
\Delta = \sqrt{2}\left[2\sin^2(\beta) - \cosh(2\beta) + 1\right]
\tag{A27}
$$

so that $\Delta < 0$ for $\beta > 0$, and $\lim_{\beta\to 0}\Delta = 0$. Therefore, the system (A20) admits a unique solution as long as $L$ is non-zero, and $L_D$ finite. Solving (A25) and substituting into (A24) leads to the solution to (A20), that can be written

$\quad \boldsymbol{c} = \mathbf{M}\boldsymbol{r}$ $\qquad\qquad\qquad\qquad\qquad\qquad\qquad\qquad\qquad\qquad\qquad\qquad\qquad\qquad\qquad\qquad$ (A28)

where the coefficients of the matrix $\mathbf{M}$ are

$$M_{11} = \frac{1}{Q}e^{-2\beta}\left[\sqrt{2}\sin\left(2\beta + \frac{\pi}{4}\right) - e^{-2\beta}\right] \tag{A29a}$$

$$M_{12} = -\frac{1}{Q}e^{-\beta}\left[\sqrt{2}\cos\left(\beta + \frac{\pi}{4}\right) + e^{-2\beta}\left(3\sin\beta - \cos\beta\right)\right] \tag{A29b}$$

$$M_{13} = \frac{1}{\tilde{\sigma}Q}e^{-2\beta}\left[\sin(2\beta) - 1 + e^{-2\beta}\right] \tag{A29c}$$

$$M_{14} = \frac{1}{\tilde{\sigma}Q}e^{-\beta}\left[\cos\beta - e^{-2\beta}\left(2\sin\beta + \cos\beta\right)\right] \tag{A29d}$$

$$M_{21} = \frac{1}{Q}e^{-2\beta}\left[-\sqrt{2}\cos\left(2\beta + \frac{\pi}{4}\right) + 2 - e^{-2\beta}\right] \tag{A29e}$$

$$M_{22} = \frac{\sqrt{2}}{Q}e^{-\beta}\left[-\sin\left(\beta + \frac{\pi}{4}\right) + e^{-2\beta}\cos\left(\beta + \frac{\pi}{4}\right)\right] \tag{A29f}$$

$$M_{23} = \frac{1}{\tilde{\sigma}Q}e^{-2\beta}\left[1 - \cos(2\beta)\right] \tag{A29g}$$

$$M_{24} = \frac{1}{\tilde{\sigma}Q}e^{-\beta}\sin\beta\left[1 - e^{-2\beta}\right] \tag{A29h}$$

$$M_{31} = \frac{1}{Q}\left[-1 + \sqrt{2}e^{-2\beta}\cos\left(2\beta + \frac{\pi}{4}\right)\right] \tag{A29i}$$

$$M_{32} = \frac{1}{Q}e^{-\beta}\left[3\sin\beta + \cos\beta - \sqrt{2}e^{-2\beta}\sin\left(\beta + \frac{\pi}{4}\right)\right] \tag{A29j}$$

$$M_{33} = \frac{1}{\tilde{\sigma}Q}\left[-1 + e^{-2\beta}\left(\sin(2\beta) + 1\right)\right] \tag{A29k}$$

$$M_{34} = \frac{1}{\tilde{\sigma}Q}e^{-\beta}\left[\cos\beta - 2\sin\beta - e^{-2\beta}\cos\beta\right] \tag{A29l}$$

$$M_{41} = \frac{1}{Q}\left[1 + e^{-2\beta}\left(\sqrt{2}\sin\left(2\beta + \frac{\pi}{4}\right) - 2\right)\right] \tag{A29m}$$

$$M_{42} = M_{22} \tag{A29n}$$

$$M_{43} = M_{23} \tag{A29o}$$

$$M_{44} = M_{24} \tag{A29p}$$

with

$$Q = -2\tilde{\sigma}^2\left[\left(1 - e^{-2\beta}\right)^2 + 2e^{-2\beta}\left(\cos(2\beta) - 1\right)\right]. \tag{A30}$$

The coefficients of $\mathbf{M}$ are implicit functions of $\tilde{\sigma}$, and we note that the coefficients $M_{1j}$ are even-symmetric to the coefficients $M_{3j}$, and the coefficients $M_{2j}$ are odd-symmetric to the coefficients $M_{4j}$. This reproduces the respective evenness and oddness of $f_1$ and $f_3$, and $f_2$ and $f_4$. The leading exponential terms for all the coefficients $M_{1j}$ and $M_{2j}$ ensure that the deflection does not diverge for large floes.

The homogeneous solution to (A1) is then

$$w_h(x) = \sum_j c_j f_j(x) \tag{A31}$$

with the coefficients of $\boldsymbol{f}$ given from (A28).

### A1.4  Summary

The solution to the BVP (A1) is given by the sum

$$w(x) = w_h(x) + w_p(x) \tag{A32a}$$

$$= \sum_{j=1}^{4} c_j f_j(x) + \sum_{j=1}^{N_f} \mathrm{Im}\left[\tilde{a}_j e^{\mathrm{i}\hat{k}_j x}\right]. \tag{A32b}$$

The definitions of the functions $f_j$ are given in Section A1.1, the definitions of the coefficients $\tilde{a}_j$ as well as the complex wavenumbers $\hat{k}_j$ are given in Section A1.2, and the definitions of the coefficients $c_j$ in Section A1.3. The integer $N_f$ is the number of frequency bins used to discretise a wave spectrum. The deflection is entirely determined by the elastic length $L_D$, the floe length $L$, and $N_f$ tuples of amplitude $a$, wavenumber $k$, attenuation number $\alpha$, and phase $\phi$. Assuming independence of these quantities, the solution is parametrised by $2 + 4N_f$ real numbers. All of these, at the exceptions of the phases taking values in $(-\pi, \pi]$, are positive. They can be further constrained to physically realistic ranges.

### A2  Elastic energy

### A2.1  Introduction

The elastic energy of a bent floe is defined as

$$E = \frac{D}{2h} \int_0^L \left(\frac{\mathrm{d}^2 w}{\mathrm{d}x^2}\right)^2 \mathrm{d}x. \tag{A33}$$

We introduce the floe curvature $\kappa(x) := \frac{\mathrm{d}^2 w}{\mathrm{d}x^2}$. From (A32), we have

$$\kappa_h = 2\tilde{\sigma}^2(-c_1 f_2 + c_2 f_1 + c_3 f_4 - c_4 f_3) \tag{A34}$$

$$\kappa_p = -\sum_j \mathrm{Im}[\hat{k}_j^2 \tilde{a}_j e^{\mathrm{i}\hat{k}x}]. \tag{A35}$$

Let us define $b_j := |\hat{k}_j^2 \tilde{a}_j|$, $\beta_j := \mathrm{Ang}\, \hat{k}_j^2 \tilde{a}_j$. We can then rewrite

$$\kappa_p = -\sum_j b_j e^{-\alpha_j x} \sin(k_j x + \beta_j). \tag{A36}$$

Finally, we introduce the quantities

$$E_h = \int_0^L {\kappa_h}^2 \, \mathrm{d}x, \; E_p = \int_0^L {\kappa_p}^2 \, \mathrm{d}x, \; E_q = \int_0^L \kappa_h \kappa_p \, \mathrm{d}x, \tag{A37}$$

which are the contribution to the elastic energy of respectively the homogeneous part of the displacement, the inhomogeneous part of the displacement, and their quadratic interaction.

 **A2.2   Homogeneous contribution**

We start by expanding $E_h$ as

$$E_h = c_1{}^2 I_2 + c_2{}^2 I_1 + c_3{}^2 I_4 + c_4{}^2 I_3 + 2(-c_1 c_2 I_{12} - c_1 c_3 I_{24} + c_1 c_4 I_{23} + c_2 c_3 I_{14} - c_2 c_4 I_{13} - c_3 c_4 I_{34}), \quad \text{(A38)}$$

with

$$I_j = \int_0^L f_j{}^2 \, \mathrm{d}x, \quad I_{jn} = \int_0^L f_j f_n \, \mathrm{d}x. \tag{A39}$$

These integrals evaluate to

$$I_1 = \frac{e^{2\beta}(\sqrt{2}\sin(2\beta + \frac{\pi}{4}) + 2) - 3}{8\tilde{\sigma}} \tag{A40a}$$

$$I_2 = \frac{e^{2\beta}(-\sqrt{2}\sin(2\beta + \frac{\pi}{4}) + 2) - 1}{8\tilde{\sigma}} \tag{A40b}$$

$$I_3 = \frac{-e^{-2\beta}(\sqrt{2}\cos(2\beta + \frac{\pi}{4}) + 2) + 3}{8\tilde{\sigma}} \tag{A40c}$$

$$I_4 = \frac{e^{-2\beta}(\sqrt{2}\cos(2\beta + \frac{\pi}{4}) - 2) + 1}{8\tilde{\sigma}} \tag{A40d}$$

$$I_{12} = \frac{-\sqrt{2}e^{2\beta}\cos(2\beta + \frac{\pi}{4}) + 1}{8\tilde{\sigma}} \tag{A40e}$$

$$I_{13} = \frac{L}{2} + \frac{\sin(2\beta)}{4\tilde{\sigma}} \tag{A40f}$$

$$I_{14} = \frac{\sin^2 \beta}{2\tilde{\sigma}} \tag{A40g}$$

$$I_{23} = \frac{\sin^2 \beta}{2\tilde{\sigma}} \tag{A40h}$$

$$I_{24} = \frac{L}{2} - \frac{\sin(2\beta)}{4\tilde{\sigma}} \tag{A40i}$$

$$I_{34} = \frac{-\sqrt{2}e^{-2\beta}\sin(2\beta + \frac{\pi}{4}) + 1}{8\tilde{\sigma}}. \tag{A40j}$$

The products of $c_j, c_n$ pairs simplify little, and can be evaluated numerically.

**A2.3   Particular contribution**

We can expand $E_p$ as

$$E_p = \sum_{j=1}^{N_f} \left[ b_j{}^2 I_j^p + 2 \sum_{n=j+1}^{N_f} b_j b_n I_{j,n} \right] \tag{A41}$$

with

$$I_j^p = \frac{1}{2} \int_0^L e^{-2\alpha_j x} \left[1 - \cos\left(2k_j x + 2\beta_j\right)\right] \mathrm{d}x \tag{A42a}$$

$$I_{j,n} = \frac{1}{2} \int_0^L e^{-\alpha_{j,n} x} \left[\cos\left(k_{j,n}^- x + \beta_{j,n}^-\right) - \cos\left(k_{j,n}^+ x + \beta_{j,n}^+\right)\right] \mathrm{d}x \tag{A42b}$$

where

$$\alpha_{j,n} := \alpha_j + \alpha_n, k_{j,n}^\pm := k_j \pm k_n, \beta_{j,n}^\pm := \beta_j \pm \beta_n. \tag{A43}$$

Let us define $K_j := |\hat{k}_j|$, $\theta_j := \mathrm{Ang}\,\hat{k}_j$. Assuming $\alpha_j \neq 0$, we can then evaluate

$$I_j^p = \frac{1}{4} \left[\frac{1 - e^{-2\alpha_j L}}{\alpha_j} + \frac{\sin(2\beta_j - \theta_j) - e^{-2\alpha_j L}\sin(2k_j L + 2\beta_j - \theta_j)}{K_j}\right]. \tag{A44}$$

Similarly, we define $K_{j,n}^\pm := |k_{j,n}^\pm + \mathrm{i}\alpha_{j,n}|$ and $\theta_{j,n}^\pm := \mathrm{Ang}\left(k_{j,n}^\pm + \mathrm{i}\alpha_{j,n}\right)$, which leads to

$$I_{j,n} = \frac{1}{2} \left[\frac{\sin(\beta_{j,n}^+ - \theta_{j,n}^+) - e^{-\alpha_{j,n} L}\sin(k_{j,n}^+ L + \beta_{j,n}^+ - \theta_{j,n}^+)}{K_{j,n}^+}\right.$$
$$\left. - \frac{\sin(\beta_{j,n}^- - \theta_{j,n}^-) - e^{-\alpha_{j,n} L}\sin(k_{j,n}^- L + \beta_{j,n}^- - \theta_{j,n}^-)}{K_{j,n}^-}\right]. \tag{A45}$$

## A2.4 Quadratic interaction contribution

We can expand $E_q$ as

$$E_q = \int_0^L -2\tilde{\sigma}^2(-c_1 f_2 + c_2 f_1 + c_3 f_4 - c_4 f_3) \sum_j \mathrm{Im}[\hat{k}_j^2 \tilde{a}_j e^{\mathrm{i}\hat{k}x}] \tag{A46}$$

$$= -2\tilde{\sigma}^2 \sum_{j=1}^{N_f} b_j[-c_1 I_{2,j}^q + c_2 I_{1,j}^q + c_3 I_{4,j}^q - c_4 I_{3,j}^q] \tag{A47}$$

with

$$I_{n,j}^p = \int_0^L f_n(x)\exp\left(-\alpha_j x\right)\sin\left(k_j x + \beta_j\right) \mathrm{d}x. \tag{A48}$$

We define

$$Q_j^{++} = (\tilde{\sigma} + \alpha_j)^2 + (\tilde{\sigma} + k_j)^2 \tag{A49a}$$

$$Q_j^{+-} = (\tilde{\sigma} + \alpha_j)^2 + (\tilde{\sigma} - k_j)^2 \tag{A49b}$$

$$Q_j^{-+} = (\tilde{\sigma} - \alpha_j)^2 + (\tilde{\sigma} + k_j)^2 \tag{A49c}$$

$$Q_j^{--} = (\tilde{\sigma} - \alpha_j)^2 + (\tilde{\sigma} - k_j)^2, \tag{A49d}$$

noting $Q^{--}$ is non-zero under the same condition (A19) that $\tilde{a}$ exists.

$$
\begin{aligned}
I_{2,j}^p = e^{-\beta} \Bigg\{ &K_j \Bigg[ \sin(\theta_j - \beta_j) \left( \frac{1}{Q^{-+}} - \frac{1}{Q^{--}} \right) \\
&+ e^{-\alpha_j L} \left[ \frac{\sin\big((\tilde{\sigma}+k_j)L - \theta_j + \beta_j\big)}{Q^{-+}} + \frac{\sin\big((\tilde{\sigma}-k_j)L + \theta_j - \beta_j\big)}{Q^{--}} \right] \Bigg] \\
&+ \sqrt{2}\tilde{\sigma} \Bigg[ -\frac{\sin\big(\beta_j + \frac{\pi}{4}\big)}{Q^{-+}} - \frac{\cos\big(\beta_j + \frac{\pi}{4}\big)}{Q^{--}} \\
&\hspace{4cm} + e^{-\alpha_j L} \left[ \frac{\sin\big((\tilde{\sigma}+k_j)L + \beta_j + \frac{\pi}{4}\big)}{Q^{-+}} - \frac{\sin\big((\tilde{\sigma}-k_j)L - \beta_j + \frac{\pi}{4}\big)}{Q^{--}} \right] \Bigg] \Bigg\} \quad \text{(A50a)}
\end{aligned}
$$

$$
\begin{aligned}
I_{1,j}^p = e^{-\beta} \Bigg\{ &K_j \Bigg[ \cos(\theta_j - \beta_j) \left( \frac{1}{Q^{-+}} + \frac{1}{Q^{--}} \right) \\
&+ e^{-\alpha_j L} \left[ -\frac{\cos\big((\tilde{\sigma}+k_j)L - \theta_j + \beta_j\big)}{Q^{-+}} + \frac{\cos\big((\tilde{\sigma}-k_j)L + \theta_j - \beta_j\big)}{Q^{--}} \right] \Bigg] \\
&+ \sqrt{2}\tilde{\sigma} \Bigg[ \frac{\cos\big(\beta_j + \frac{\pi}{4}\big)}{Q^{-+}} - \frac{\sin\big(\beta_j + \frac{\pi}{4}\big)}{Q^{--}} \\
&\hspace{4cm} + e^{-\alpha_j L} \left[ -\frac{\cos\big((\tilde{\sigma}+k_j)L + \beta_j + \frac{\pi}{4}\big)}{Q^{-+}} + \frac{\cos\big((\tilde{\sigma}-k_j)L - \beta_j + \frac{\pi}{4}\big)}{Q^{--}} \right] \Bigg] \Bigg\} \quad \text{(A50b)}
\end{aligned}
$$

$$
\begin{aligned}
I_{4,j}^p = e^{-\beta} \Bigg\{ &K_j \Bigg[ -\sin(\theta_j - \beta_j) \left( \frac{1}{Q^{++}} - \frac{1}{Q^{+-}} \right) \\
&+ e^{-\alpha_j L} \left[ -\frac{\sin\big((\tilde{\sigma}+k_j)L - \theta_j + \beta_j\big)}{Q^{++}} + \frac{\sin\big((\tilde{\sigma}-k_j)L + \theta_j - \beta_j\big)}{Q^{+-}} \right] \Bigg] \\
&+ \sqrt{2}\tilde{\sigma} \Bigg[ -\frac{\cos\big(\beta_j + \frac{\pi}{4}\big)}{Q^{++}} - \frac{\sin\big(\beta_j + \frac{\pi}{4}\big)}{Q^{+-}} \\
&\hspace{4cm} + e^{-\alpha_j L} \left[ \frac{\cos\big((\tilde{\sigma}+k_j)L + \beta_j + \frac{\pi}{4}\big)}{Q^{++}} - \frac{\cos\big((\tilde{\sigma}-k_j)L - \beta_j + \frac{\pi}{4}\big)}{Q^{+-}} \right] \Bigg] \Bigg\} \quad \text{(A50c)}
\end{aligned}
$$

$$
\begin{aligned}
I_{3,j}^p = e^{-\beta} \Bigg\{ &K_j \Bigg[ -\cos(\theta_j - \beta_j) \left( \frac{1}{Q^{++}} - \frac{1}{Q^{+-}} \right) \\
&+ e^{-\alpha_j L} \left[ \frac{\cos\big((\tilde{\sigma}+k_j)L - \theta_j + \beta_j\big)}{Q^{++}} + \frac{\cos\big((\tilde{\sigma}-k_j)L + \theta_j - \beta_j\big)}{Q^{+-}} \right] \Bigg] \\
&+ \sqrt{2}\tilde{\sigma} \Bigg[ -\frac{\sin\big(\beta_j + \frac{\pi}{4}\big)}{Q^{++}} - \frac{\cos\big(\beta_j + \frac{\pi}{4}\big)}{Q^{+-}} \\
&\hspace{4cm} + e^{-\alpha_j L} \left[ \frac{\sin\big((\tilde{\sigma}+k_j)L + \beta_j + \frac{\pi}{4}\big)}{Q^{++}} - \frac{\sin\big((\tilde{\sigma}-k_j)L - \beta_j + \frac{\pi}{4}\big)}{Q^{+-}} \right] \Bigg] \Bigg\}. \quad \text{(A50d)}
\end{aligned}
$$

## Appendix B: Comparison to another mechanical model

In this Section, we compare the solution to floe bending issued from the model described in this publication, SWIIFT, to results issued from a model that solves for wave scattering from ice floes, hereafter referred to as WISIB (Mokus and Montiel, 2022). The main difference is that in the latter case, floe deflection is derived from the interface between fluid and floe, assuming that the floe conforms exactly to the fluid; the solution is sought assuming harmonic forcing of the plate, and incorporates forward and backward travelling waves, while SWIIFT only represents forward travelling waves. A consequence is the possibility for WISIB to create constructive interference locally increasing the deformation of the plate or its curvature. Interactions between floe lengths and wavenumbers can locally lead to resonances.

In Sect. B1, we look at curvature envelopes. In Sect. B2, we look at potential elastic energy derived from these curvatures.

### B1  Curvature envelopes

In Fig. B1, we compare curvature envelopes derived from SWIIFT or WISIB. We do so for various ice thicknesses, wave periods, and floe lengths. By curvature envelope, we mean the maximum curvature attainable at any location along the length of a floe; the actual curvature oscillates and reaches it at its positive antinodes. In the case of WISIB, curvature is the sum of forward and backward travelling modes, forward and backward damped modes, and evanescent modes. As a consequence, the envelope itself oscillates. When the wavelength is long enough compared to the floe, these oscillations disappear.

When the wavelength gets significantly longer than the floe (for example, $T = 10\,\mathrm{s}$ and $12\,\mathrm{s}$ and $h = 25\,\mathrm{cm}$, most floe lengths), the curvature envelopes are different near the edges: SWIIFT has damped terms which oscillates with spatial frequency $\frac{1}{\sqrt{2}L_D}$, while WISIB has damped terms which are complex solutions to the dispersion relation Eq. (19). The former depends only on ice thickness, while the latter depends primarily on the wave period.

### B2  Energy from spectral forcing

More than the curvature itself, what matters to our energy-based fracture parametrisation is the potential elastic energy of a deformed floe. To compare these energies between the two models, we calculate them from the curvatures derived from both mechanical models. We do so for a spectral forcing, corresponding to a Pierson–Moskowitz spectrum discretised onto 47 frequency bins, between 0.05 Hz to 0.52 Hz, with a width of 1 Hz. We use Latin hypercube sampling to generate an ensemble (size 289) of ice thicknesses, significant wave heights (parametrising the spectrum), and floe lengths, from uniform distributions on respectively 25 cm to 100 cm, 2 m to 4 m, and 50 m to 400 m. The spectra, integrated on the discretised frequency axis, show a median relative error to the targets $\frac{H_S^2}{16}$ of $3.3 \times 10^{-3}$. To each frequency bin, we associate a phase randomly sampled from 0 to $2\pi$, to build an incoherent wave field, from which floe curvature, and eventually, elastic energy, is derived. We show the results in Fig. B2.

The resulting energies show a dependency to each of the three chosen variables, highlighted by the regression lines on the top row. The energy derived from SWIIFT is generally higher than the energy derived from WISIB, and exhibit more spread. However, the trends are similar for the energies derived from both models, so that the ratio of the two does not show any

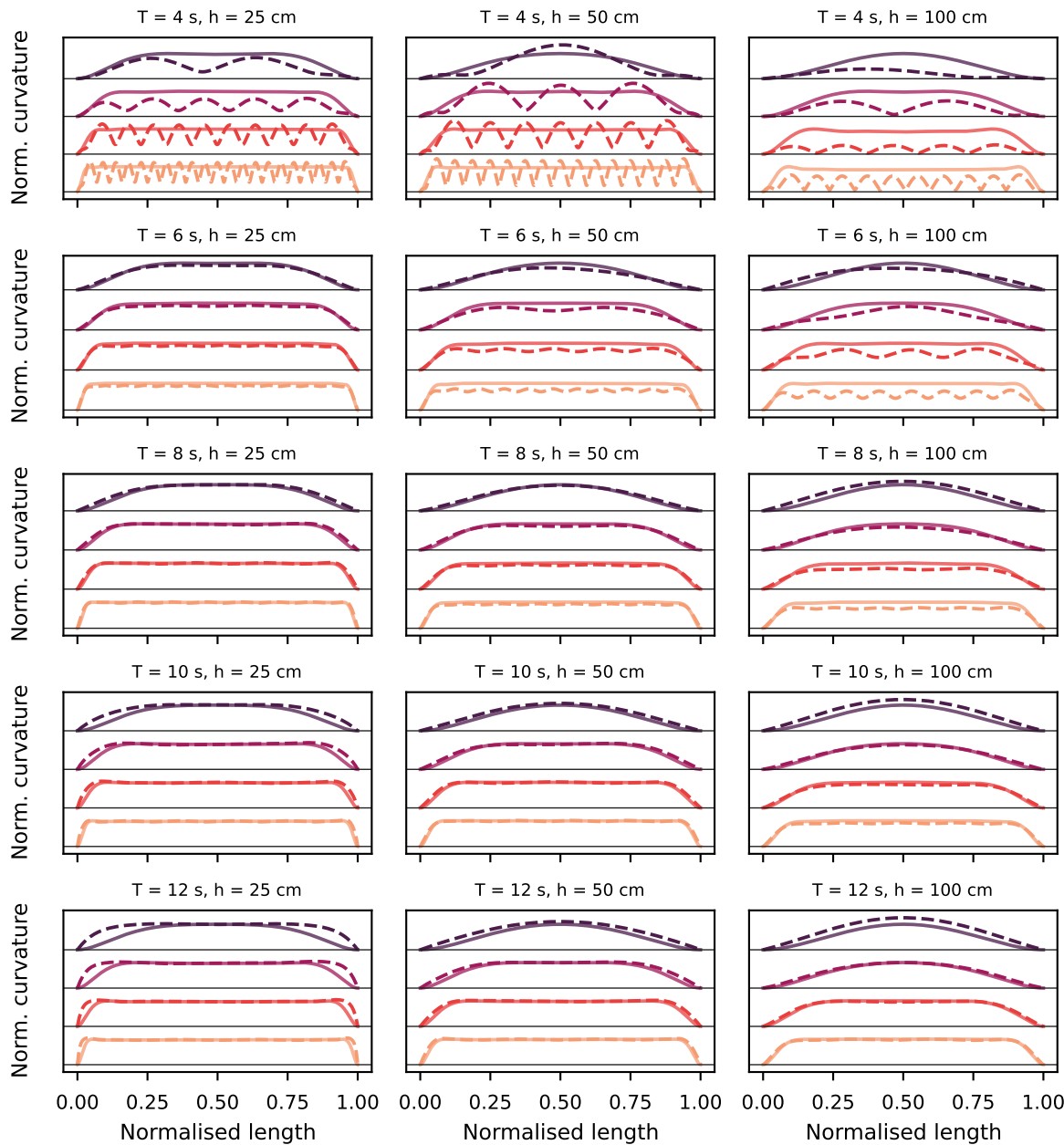

**Figure B1.** Comparison of curvature envelopes derived from SWIIFT or WISIB for different periods of wave forcing (row) and different ice thicknesses (columns). Within each panel, from top to bottom (and darker to lighter hue) are floe lengths of $50\,\text{m}$, $100\,\text{m}$, $200\,\text{m}$ and $400\,\text{m}$, and the solid line is the SWIIFT solution, while the dashed line is the WISIB solution. The x-axes are normalised with respect to floe lengths, and individual curvatures are normalised with respect to the maximal curvature computed with SWIIFT. We display only the positive branch of the envelopes, which are symmetrical with respect to each corresponding y-axis. The origin of these y-axes is shown with a thin horizontal black line.

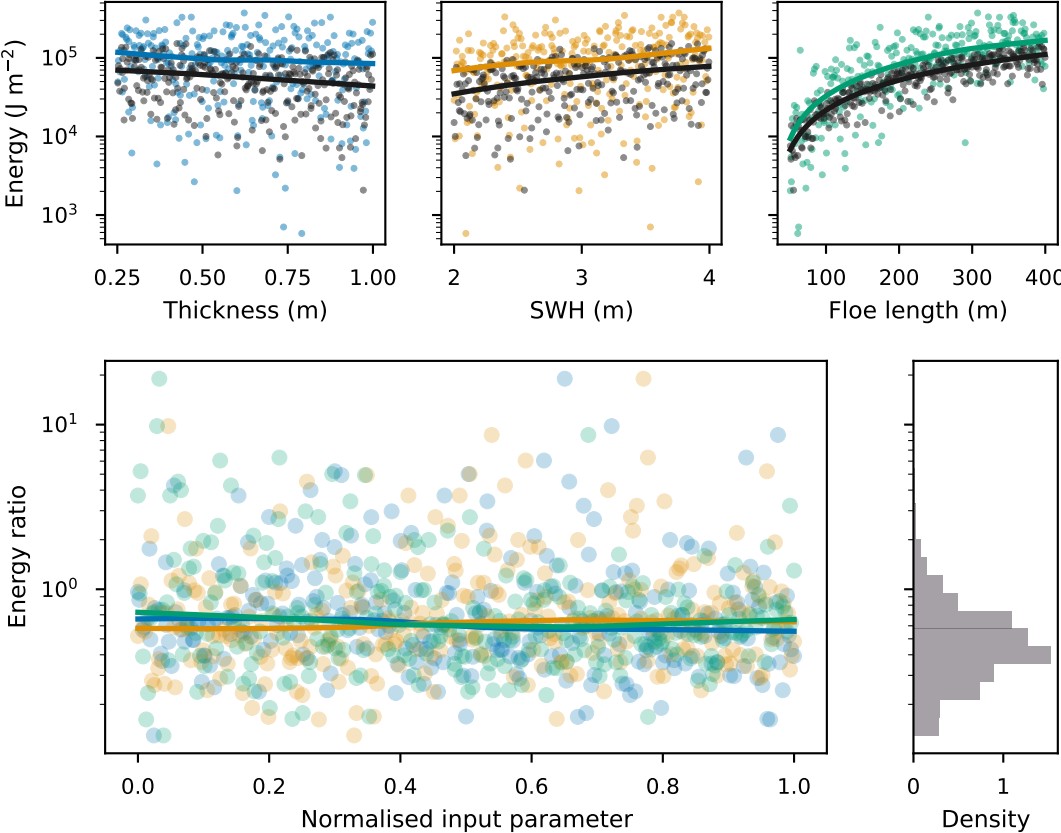

**Figure B2.** Comparison of potential elastic energy derived from SWIIFT or from WISIB. Top row: energy as a function of three parameters, with coloured dots for SWIIFT and black dots for WISIB. Lowess regression lines are superimposed. Bottom row: on the right, ratios of energy, as described in the text; the colours correspond to the variables of the top row, which were normalised to fit on the same axis. On the right, distribution of the energy ratio.

dependency to either of the variables (the regression lines on bottom left panel of Fig. B2 are mostly horizontal). In the bottom right panel of Fig. B2, we show the distribution of these ratios. It is right-skewed, and has mean $1.02$ (geometric mean $0.69$).

In Fig. B3, we show the correlation between our three input variables and the resulting energies. Because the trends exhibited on the top panel of Fig. B2 are non-linear, we use the Spearman correlation coefficient, which quantifies the monotonicity of a relationship. As can be seen from Fig. B2, and particularly the regression lines, the energy computed from either model are very mildly negatively correlated with thickness, mildly positively correlated with significant wave height, and clearly positively correlated with floe length. The correlations are stronger when using WISIB, which produces less scattered results. The energy

ratio, however, is at most very lightly correlated with any of the variables.

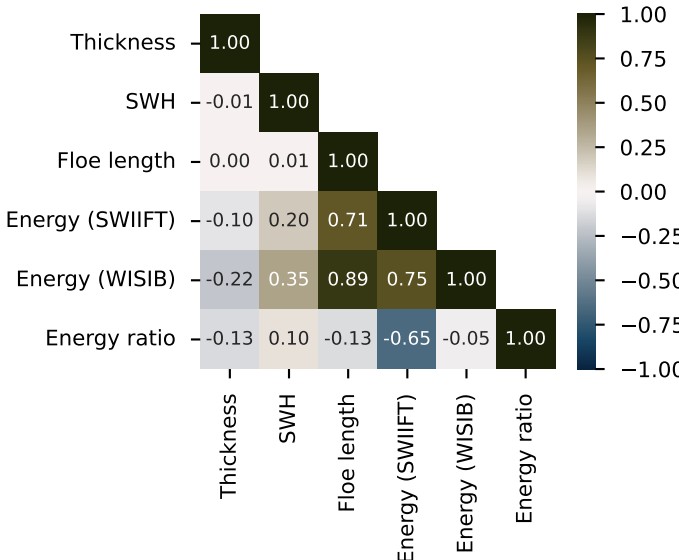

**Figure B3.** Correlation matrix between input parameters and energy derived from SWIIFT and WISIB. We use the Spearman correlation coefficient.

*Author contributions.* NM developed the model off an initial version developed by JPA and AT, designed the numerical experiments, and conducted the analysis, with input from the other authors. NM and VD led the writing with suggestions and improvements from GB. VD supervised the study.

*Competing interests.* The authors have no competing interests.

*Acknowledgements.* The authors thank Baptiste Auvity, Stéphane Perrard, Antonin Eddi, Dany Dumont, and Vasco Zanchi for fruitful discussions which have significantly improved the quality of this manuscript.

*Financial support.* This research received support through Schmidt Sciences, LLC, via the SASIP project (grant G-24-67788).

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
