# Peer review of "SWIIFT v0.10: a numerical model of wave-induced sea ice breakup with an energy criterion"

_EGUsphere, 2025_

## Author Comment (AC1)

**Response to RC1**

**1 Opening statement**

*SWIIFT v0.10 presents an algorithm to predict wave-induced breakup of sea ice floes. Contrary to the most widespread assumption of a critical strain for breakup, the authors implement a different physical mechanism based on energy. I do feel that the model is appropriately described and assumptions are physically justified making it very valuable contribution to the active field of research on waves and sea ice, however some statements need to be recalibrated in view of recent literature, which in parts is omitted, and to avoid overstating the value of the present contribution.*

We thank you for your thorough review of our manuscript and this positive foreword. The purpose of our paper is to present an alternative modelling approach to that currently existing and applied in floe-resolving models, as well as to perform a first validation of this approach based on a comparison to experimental data. In particular, our goal is not to invalidate existing frameworks. Additionally, our focus is on a single process: the mechanical fracture of sea ice in a brittle, cohesive solid state under wave action. We purposely exclude thermodynamics and failure of ice under other states and processes from our study. Thanks to your useful review, we have become aware that this scope and more precise objectives were not emphasised enough in our introduction and we have now stated it more clearly. We provide detailed responses to your individual comments hereafter.

**2 Major comments**

**2.1**

*The authors focus on wave induced breakup. This is one of the possible mechanisms leading to the formation of the MIZ but not the only one, and this should be made clearer in the abstract and*

*introduction. For example, internal stresses can be induced by wind and current forcing, and the weakening of the ice cover that promotes breakup to thermodynamic effects (e.g. melting). Moreover, to my understanding, the paper focuses on the condition in which the floes are comparable to the wavelength. While I appreciate that in this condition waves 'build' the MIZ via breakup, this is only true in particular seasons and locations. The authors overlook the formation of the MIZ via for example the pancake ice cycle (in which floes much smaller than the wavelength) and is linked both to the agitation induced by the waves (mechanical process) and thermodynamic freezing.*

In the abstract (first sentence), we state that 'The wave-induced breakup of sea ice *contributes* to the formation of the marginal ice zone in the polar oceans' (emphasis added), suggesting we do not assert wave-induced breakup is the only contributor to the MIZ. Early in our introduction, we list wind and ocean currents as mechanisms of fragmentation.

It is true that the focus of our introduction then shifts to wave-related matters, as the purpose of our paper is to present a method for parametrising the flexural failure of cohesive, solid sea ice under wave action. A mechanism, as you say, that contributes to building the MIZ, and which produces ice floes of sizes comparable to (but largely smaller than) the wavelength, starting from larger, unbroken floes. We now emphasise this point in the revised version of our introduction. We also explicitly state that our model is purely mechanical and purposely does not incorporate any thermodynamics—the omission of that clarification was an oversight on our part—and that, as a consequence, we do not represent any kind of ice formation, melt, or disintegration.

We also now make an explicit distinction in the introduction between the Marginal Ice Zone (MIZ), qualitatively defined as the area influenced by wave action, and the Seasonal Ice Zone (SIZ, see for example Roach et al. (2025)), qualitatively defined by opposition to perennial ice. The floe size distribution (FSD) can also be studied at the scale of the SIZ, and we focus here solely on one process: wave-induced breakup.

**2.2**

*One of the claims, as highlighted in the abstract, is that maximum strain might not be the dominant mechanism. While the energetic criterion proposed might be physically sound, a more throughout comparison with different breaking modes as discussed in a re-*

*cent paper by Saddier et al (`https://journals.aps.org/prfluids/abstract/10.1103/PhysRevFluids.9.094302?ft=1`) should have been considered. Moreover, the calling in the question the maximum strain criterion is not completely novel. For example, in Passerotti et al, that the authors discuss, it was already shown that existing criteria do not match experimental observations.*

We thank you for your comment but respectfully disagree on two aspects. First, we did not claim in our paper that maximum strain is not the dominant mechanism for flexural wave breaking. Rather, we presented a common paradigm (breaking parametrised with a strain threshold criterion) to introduce an alternative paradigm (breaking parametrised with an energy criterion). We take note to emphasise in the introduction that maximum strain is not a mechanism, but a criterion that can be used to parametrise a mechanism: wave-induced fracture. We actually gave examples of the strain parametrisations agreeing with observations and being used in models of various scales (line 45 of our original manuscript). In any case, both the fracture criteria we discuss are built around bending strain: the strain threshold method is a local comparison, and the energy method integrates its variation along the plate.

Second, we did not claim that calling the maximum strain criterion into question was novel: the results we present in this study draw heavily from the work of Auvity et al., under review for PRL, which concluded that this criterion was inconsistent with their experimental results. This is clearly stated from line 49 of our manuscript. Passerotti et al. (2022) compared their experimental results to the so-called universal criterion proposed by Voermans et al. (2020), and found the match not to be perfect, with some fractures observed below the threshold and some absence of fracture beyond the threshold. They were careful to frame this finding in the appropriate context of their physical setup, but did not discuss the possible effect of material fatigue: they use a single ice sheet across their experiments. The criterion of Voermans et al. (2020), however, already is a blend of wave properties and ice properties, and not just the value of a strain threshold, and they do not discuss whether considering a strain threshold in isolation is appropriate or not. However, we did look at the literature again, and could not find work clearly calling into question the maximum strain criterion framework.

Saddier et al. (2024) suggested the fracture they observed came from viscous stress and acknowledged this is not the mechanism that leads to the fracture of ice floe by waves. Their material is held together by capillarity, and is much thinner than the viscous boundary layer associated with

their flow. Auvity et al. (2025) identified the failure of their material to be caused by wave-induced bending, as it is the case for ice floes. Therefore, we focus exclusively on mode I for the simple reason that our geometry is one-dimensional: it does not allow for representing anything else.

**2.3**

*The authors make a thorough comparison to the experiments of Auvity, a preprint. The experiments are done for a standing wave, which is an unlikely condition to be observed in the ocean where waves are likely to propagate from the open ocean towards the sea ice. I wonder why a greater effort has not been made to make a comparison to laboratory experiments of Passerotti that the author mentions (noting that these encompass a more complex random sea state). Moreover, striking is the absence in their work of mention to the work of Saddier et al that, in my view, closely resembles the one of Auvity, albeit with few notable differences (e.g. propagating waves vs standing waves, and also random waves). In addition, I feel that the authors oversell the model agreement with the experiments (Fig 8).*

In this line of thoughts, simplicity here is sought. We referenced the experiment of Passerotti et al. (2022) (line 520 in the original manuscript) and explained why it was not considered for comparison. We aimed to look at the smallest interesting problem involving fracturing, since we seek to validate the alternative energetic approach to fracturing implemented in our model; hence monochromatic forcing, no attenuation, and standing waves. We now make this point clearer in the introduction. As described in the paper, the model, in its present state, can do more complex things. As you pointed it out in your next comment, it would be relevant to demonstrate this capacity: we therefore now include a propagative example in an additional section.

However, the point of this paper is, first, to introduce the software and the bases it rests upon and, second, to make sure that it is relevant and its results sensible in a simple case. One of our conclusion, as stated in the paper, is that even in this simplest case, the results are not straightforward to interpret.

Studying standing waves makes it possible to calculate a threshold amplitude, a physical quantity measured by experimenters, allowing for direct comparison. We acknowledge that the threshold amplitude of a transient forcing could be different, as reported by Saddier et al. (2024). The presented

setup allows for deriving results confirming that our breakup criterion, if not perfect, is not completely off either: and we think the comparison to Auvity et al. (2025) is the most relevant for doing so. We do not think we oversold the agreement: for example, text line 411–414, line 482–486, we insisted on having an agreement in terms of order of magnitude. We also do not tune the model parameters to obtain this agreement, an information that, as you point out in your later comment, was not made clear enough in the original version of our manuscript.

Even though similar in principle, the main difference between the works of Saddier et al. (2024) and Auvity et al. (2025) is that in the former case, the material floats because of capillarity and breaks because of viscous stress; in the latter cases, it floats because of buoyancy, and breaks under bending stress, a situation much more similar to what happens to ice floes, and that our model tries to emulate. Not mentioning this work was, however, clearly an oversight on our part, and we amended the introduction of our Section 3 to fix it.

**2.4**

*As a further suggestion, I believe that a working example with propagating ocean waves and a random sea state could be added to the manuscript and it would strengthen the paper.*

We thank you for the suggestion. We added Section 2.4.5 to present such an example, illustrated in Fig. 1.

**3 Additional comments**

*Additional detailed comments are listed below.*

**3.1**

*In their modelling paradigm, the energy release rate G is introduced. Can the authors please explain and or suggest how its value can be evaluated in the field and lab experiments. Otherwise, this remains as a fitting parameter.*

This parameter can be measured in the laboratory or on the field, either directly, or from the fracture toughness to which it is related through the Young's modulus, for example by three-point bending tests. Our original manuscript alluded to it in the introduction (line 59), where we now make this

[Figure]

Figure 1: Snapshots of a fracture experiment. Top panel, view of the domain at $t = 60$ s. The continuous, dark line represents the fluid surface $(\eta(x))$, and the discontinuous, lighter lines the vertical displacements $(w(x))$ of individual floes. The marks along the bottom spine indicate the boundaries between fragments; about 80 m at the right of the domain have not yet been affected by the waves. Note that the vertical scale is greatly exaggerated: the aspect ratio of the graph, in physical units, is $5 \times 10^{-4}$. Because of the thickness of the lines, some floes appear to overlap, they actually do not. Bottom left panel, horizontal bars show the extent of individual floes. The height of the bars indicates the order of the floe in the array, and each group of bars, or "stair", corresponds to a snapshot. The time of the snapshots are indicated on the y-axis, and darker colours correspond to later times. Bottom right panel, we show size distributions as swarmplots, omitting the rightmost fragment. Each dot corresponds to a length as indicated by the x-axis, and within a group, the y-axis only serves to separate dots. Vertical clusters thus indicate a concentration of observations around the corresponding length. From $t = 0$ s to 120 s, there are respectively 1, 6, 27, 52, 58, 59 and 60 fragments.

point clearer. As it usually is the case, the mechanical properties of sea ice are less well constrained than that of other more standard material, or even fresh water ice, and can be expected to depend on temperature and brine volume fraction, and more generally on the history of the material. Timco and Weeks (2010) compiled previous studies of fracture toughness measurements. Wei and Dai (2021) conducted such measurements more recently, at the lab scale, and compared dry and wet samples. We added Section 2.4.4 to inform the reader on the values this parameter (as well as critical strain) can take.

When it comes to the results we present in Section 4, ice is not the material under consideration, and estimation of the energy release rate was done by Auvity et al. (2025). We use this estimate to parametrise all our presented results; that is, we do not tune it to adjust our results.

**3.2**

*The numerical experiments are done with a brittle layer of varnish (L268), I wonder if the hypothesis of elastic plate applies to a material that the authors define brittle.*

In their manuscript, Auvity et al. (2025) do establish the material behaves as a solid. We oppose brittle to ductile, not to elastic; that is, the material can deform, but will fracture before exhibiting significant plastic (irreversible) deformation.

**3.3**

*2.1 there are a couple of hypotheses in the modelling framework that, in my opinion, should be better highlighted. The plate is elastic (also the coefficients are those for a quasi-static model) and the ice does not drift.*

We thank you for your comment. Indeed, we consider an elastic material: no time-dependent energy dissipation occurs, other than released by fracture. That is, the material is not viscous. Dissipation is parametrised in space.

We modified the manuscript to emphasise the elasticity hypothesis (second paragraph of Section 2.1) and the fact we do not consider drift (new paragraph at the end of Section 2.1).

**3.4**

*2.3.2 the attenuation is parameterized as in Sutherland (eq 20). Can the author better justify this modelling choice and explain*

*why other approaches have not been considered. For example an emerging trend is the ones in DeSanti et al and Yu et al (https://agupubs.onlinelibrary.wiley.com/doi/full/10.1029/2018JC013865; https://www.sciencedirect.com/science/article/pii/S0165232X2200101X). Can the author please explain/comment on how different attenuation might affect their results.*

This attenuation scheme was chosen simply because one of the authors was familiar with it. In fact, in its present state (v0.10), the model can accept any user-defined function as an attenuation parameterisation, as long as it relies on the quantity numerically represented (including, but not limited to, ice thickness or wavenumber). This capability was mentioned in the original manuscript, line 214. Moreover, the code is flexible enough that other parametrisations can be offered permanently. We modified our manuscript with the hope to make this clearer.

Yu et al. (2022) relates a nondimensionalised attenuation to a nondimensionalised angular frequency, so that

$$\alpha h = 0.108 \left( \omega \sqrt{\frac{h}{g}} \right)^{4.46} \tag{1}$$

or equivalently,

$$\alpha = 0.108 \omega^{4.46} h^{1.23} g^{-2.23}. \tag{2}$$

This is close to the parametrisation we suggested, which is approximately (when making a deep water, free surface substitution for the wavenumber) $\alpha = \frac{1}{4} h \omega^4 g^{-1}$, albeit with a smaller prefactor. We added the parametrisation from Yu et al. (2022) to SWIIFT v0.16.

The approach of De Santi et al. (2018) is focused on grease ice and pancake ice, while we consider discrete floes, long enough to be susceptible to fail from bending. Additionally, the models they consider depend at the very least on the viscosity of the ice, and eventually on the viscosity of the fluid and a pancake fraction parameter. These are in opposition with our current formulation, purely elastic ice and inviscid fluid. While the model would be amendable to accommodate these, we do not deem it to be a priority. It is, however, free and open source, and contributions are welcome.

As to the eventual impact of the selection of an attenuation scheme on the results, we can for now simply say that higher attenuation would lead to a smaller extent of fracture. Additionally, we would like to again attract the attention of the readers on the fact that the results presented in Section 4 are issued of simulations where attenuation was turned off.

**3.5**

*2.3.3 I do not understand the opening statement. This is reinforced by the choice of the authors of choosing a wave expressed as a variable of the x, whereas in ocean wave applications the more common approach is to provide a time series at the edge of the domain and let it evolve along the x coordinate.*

We do not fully understand this comment, or what is the difference you mean to convey between 'a wave expressed as a variable of the x' and 'a time series at the edge of the domain [...] evolve along the x coordinate'. Dropping attenuation and superposition in the interest of simplicity, we do express waves as $\eta(x,t) = a \sin(kx - \omega t + \phi_0)$, so that the surface of the fluid at $x = 0, t = 0$ sits at $a \sin(\phi_0)$. As we consider a succession of quasi-static states, we eliminate the explicit time dependency by aggregating it into the phase, so that $\eta_n(x) := \eta(x, t = t_n) = a \sin(kx + \phi_n)$, with $\phi_n = \phi_0 - \omega t_n$. In the original manuscript, this is detailed from Section 2.3.2 to Section 2.4.1, inclusive. Additionally, to ease further computations, $x = 0$ can be chosen relatively to any floe of the domain. Thus, what we do is, in a way, precisely providing a time series at the edge of the domain (in a matter of fact, the edge of any floe) and letting it evolve along the $x$ coordinate (along the considered floe). The time information is simply encapsulated into the phase of the complex amplitude.

In Section 2.3.3, we explicit how we modify $\eta$ to allow the fluid surface to transition from a rest state (in the vicinity of the ice cover) to a 'wavy' state, in order to be able to simulate the progression of a fracture front. Equations (23) and (24) of the original manuscript are the translation of this opening sentence in mathematical terms. What we do here is simply providing a Gaussian envelope to our plane wave to locally reduce its amplitude. However, we are not interested in the progression of a single wave packet, so we only impose this envelope in a half-plane, allowing for the transition from rest to 'fully developed' sea in a continuous and regular manner.

**3.6**

*3.2 The authors make the assumption of linearity. There is no discussion on the possible effect of capillarity. In the wave regime explored in the paper (small wavelength) capillarity effect might affect the wave dispersion relation.*

Auvity et al. (2025) establish in their manuscript that in the case of their material, elastic effects completely dominate. They do so experimentally,

and it can be understood from the flexural length to capillary length ratio (about three), and their respective powers in the dispersion relation. This applies for all wavenumbers, and can be easily verified analytically. On the contrary, in the experimental conditions of Saddier et al. (2024), the flexural length to capillary length ratio is about 0.013.

As explained in Section 2.3.1, we use the gravity–mass-loading–elastic dispersion relation for our ice-covered regions. It can be extended with a compression term (Liu and Mollo-Christensen 1988), a term analogous to surface tension for a fluid–fluid interface. This term is, however, poorly constrained (Collins et al. 2017) and can be minor (Sutherland and Dumont 2018). In the case of sea ice, it is likely that the 'appropriate' dispersion relation depends on the type of ice considered, and its spatial scale, or the spatial scale of the floes.

We added a paragraph to Section 3.1, to make clear why we do not consider capillarity.

**3.7**

*Fig 4 the kL axis only spans one order of magnitude and I wonder if the log scale is really needed. Moreover, in the discussion the authors state that they only look at the plate between 0:L/2 because of symmetries. When a breakup occurs how do the authors make sure that this is in the first half of the plate and not in the second half? Is there a reason to believe that the floe breaks synchronously at two points (one in 0:L/2 and one in L/2:L) therefore forming 3 smaller floes.*

Even though $kL$ only spans one order of magnitude, we want to expose a power relationship in panel (b), which is more easily done by using a log–log scale.

In the text (line 341 of the original manuscript), we explain that the free energy profile is symmetric. Therefore, if we consider binary fracture, there exists two identical free energy minima (or a single one in the exact middle of the plate). Either one can be chosen as the 'true' fracture location, as what we quantify here is the amplitude of fracture onset. They cannot be distinguished and the only reason our minimisation step would consider one over the other, is numerical fluctuations on the order of floating point precision. We simply restrict our analysis by showing, in Figure 4a (original manuscript), fracture locations constrained to $[0, \frac{L}{2}]$. For completeness, we could have added points symmetrical to those represented with respect to $y = \frac{1}{2}$, but it would have made reading the graph harder without adding any

information.

In their experiment, Auvity et al. (2025) observed fracture on wave crests one at a time. As their goal is to identify the threshold amplitude leading to fracture, the experiments were stopped as soon as a first fracture had appeared. In the time interval necessary for stopping the experiment, secondary fractures usually appeared, not unlike what Saddier et al. (2024) observed.

**3.8**

*L21 I feel that in addition to the reference to Auclair there is observational evidence showing that the marginal ice zone affected by waves is close to free drift regime and therefore substantially different from the interior. Addition of appropriate references would strengthen the statement. Moreover, in addition to reference to Thomson, I suggest adding the recent work by Toyota et al (https: //www.sciencedirect.com/science/article/pii/S1873965225000520).*

We thank you for this suggestion. We expanded this paragraph of our introduction with references to recent observations of ice motion, and the suggested reference to the study of Toyota et al. (2025), which was not published when we submitted our manuscript.

**3.9**

*L35 I find this sentence unclear.*

Our response here refers to Figure 2 of the original manuscript. The shaded areas represent intervals, over the floe lengths, where the critical strain is exceeded, for the typically considered $\varepsilon_{\mathrm{cr}} = 3 \times 10^{-5}$. These cover almost the entire span of the floe, except for some small regions around edges and curvature nodes. Therefore, simply looking for where the critical strain is reached is not sufficient, one also has to devise a way to select where to break the floe within these intervals. The two more obvious options are to fracture at the first point where the threshold is reached, or to choose the global or a local extremum. We choose the latter, which is explicitly stated in Section 2.2.2. Other methods could be suggested. Horvat and Tziperman (2015), for example, chose not to compute the contiguous strain along a floe, but a local approximation, considering only successive extrema of the sea surface realisation (their analogue to our vertical displacement; see their supplementary material). They did so to address that very same limitation of strain-derived fracture parameterisation. To quote these authors: 'If

the strain is calculated locally from $\eta(x)$, the critical strain is reached almost everywhere for a realistically generated wave field (see the Supplement, Fig. S10)'.

**3.10**

*L137 for the readership benefit, can the author state what it means unstretchable.*

Thank you for your comment. We mean that the plate does not undergo any in-plane deformation. We added a clarification to the manuscript.

**3.11**

*L255 can the value of Y and nu be explicitly specified?*

The values reported in the cited study are $Y = 3.8\,\mathrm{GPa}$ and $\nu = 0.33$. However, we used $Y = 6\,\mathrm{GPa}$ and $\nu = 0.3$ to compute the values presented in our manuscript.

We added a clarification on the values used, and updated the results of our calculation to use these. We insist on the fact that this calibration is illustrative, and will need to be adjusted depending on the configuration of a model run.

**3.12**

*L264 the relationship for polychromatic cases should be explicitly stated for clarity.*

We made this addition to the manuscript.

**3.13**

*L420 can the author better clarify why the definition of the relaxation length differs from Auvity. Can the two be reconciled?*

The clarification is given in the following paragraph. The definition of Auvity et al. is based on the full width at half maximum. It presupposes the location of fracture is were the curvature maximum was before fracture happened, which is expected for bending failure. However, it would not be compatible with the behaviour we observe in region 4, were fracture happens away from deflection and curvature crests.

**3.14**

*L515 the example does not refer to "typical field conditions" as this is a transient ship wake and not a MIZ formed by open ocean waves.*

We replaced 'typical field conditions' by 'field scale', which is what we meant, by opposition to lab scale.

**References**

Auvity, B., L. Duchemin, A. Eddi, and S. Perrard (2025). *Wave induced fracture of a sea ice analog*. DOI: 10.48550/ARXIV.2501.04824. arXiv: 2501.04824 [physics.flu-dyn].

Collins, C. O., W. E. Rogers, and B. Lund (2017). "An investigation into the dispersion of ocean surface waves in sea ice". In: *Ocean Dynamics* 67.2, pp. 263–280.

De Santi, F., G. De Carolis, P. Olla, M. Doble, S. Cheng, H. H. Shen, P. Wadhams, and J. Thomson (Aug. 2018). "On the Ocean Wave Attenuation Rate in Grease-Pancake Ice, a Comparison of Viscous Layer Propagation Models With Field Data". In: *Journal of Geophysical Research: Oceans* 123.8, pp. 5933–5948. ISSN: 2169-9291. DOI: 10.1029/2018jc013865.

Horvat, C. and E. Tziperman (2015). "A prognostic model of the sea-ice floe size and thickness distribution". In: *The Cryosphere* 9.6, pp. 2119–2134. DOI: 10.5194/tc-9-2119-2015.

Liu, A. and E. Mollo-Christensen (1988). "Wave propagation in a solid ice pack". In: *Journal of Physical Oceanograpy* 18, pp. 1702–1712.

Passerotti, G., L. G. Bennetts, F. von Bock und Polach, A. Alberello, O. Puolakka, A. Dolatshah, J. Monbaliu, and A. Toffoli (2022). "Interactions between irregular wave fields and sea ice: A physical model for wave attenuation and ice breakup in an ice tank". In: *Journal of Physical Oceanography* 52.7, pp. 1431–1446.

Roach, L. A., M. M. Smith, A. Herman, and D. Ringeisen (2025). "Physics of the Seasonal Sea Ice Zone". In: *Annual Review of Marine Science* 17.Volume 17, 2025, pp. 355–379. ISSN: 1941-0611. DOI: https://doi.org/10.1146/annurev-marine-121422-015323.

Saddier, L., A. Palotai, M. Aksil, M. Tsamados, and M. Berhanu (Sept. 2024). "Breaking of a floating particle raft by water waves". In: *Physical Review Fluids* 9.9. ISSN: 2469-990X. DOI: 10.1103/physrevfluids.9.094302.

Sutherland, P. and D. Dumont (2018). "Marginal ice zone thickness and extent due to wave radiation stress". In: *Journal of Physical Oceanography* 48.8, pp. 1885–1901.

Timco, G. W. and W. F. Weeks (2010). "A review of the engineering properties of sea ice". In: *Cold Regions Science and Technology* 60.2, pp. 107–129. ISSN: 0165-232X. DOI: `https://doi.org/10.1016/j.coldregions.2009.10.003`.

Toyota, T., Y. Arihara, T. Waseda, M. Ito, and J. Nishioka (May 2025). "Melting processes of the marginal ice zone inferred from floe size distributions measured with a drone in the southern Sea of Okhotsk". In: *Polar Science*, p. 101215. ISSN: 1873-9652. DOI: `10.1016/j.polar.2025.101215`.

Voermans, J. J., J. Rabault, K. Filchuk, I. Ryzhov, P. Heil, A. Marchenko, C. O. Collins III, M. Dabboor, G. Sutherland, and A. V. Babanin (2020). "Experimental evidence for a universal threshold characterizing wave-induced sea ice break-up". In: *The Cryosphere* 14.11, pp. 4265–4278. DOI: `https://doi.org/10.5194/tc-14-4265-2020`.

Wei, M. and F. Dai (Aug. 2021). "Laboratory-scale mixed-mode I/II fracture tests on columnar saline ice". In: *Theoretical and Applied Fracture Mechanics* 114, p. 102982. ISSN: 0167-8442. DOI: `10.1016/j.tafmec.2021.102982`.

Yu, J., W. E. Rogers, and D. W. Wang (2022). "A new method for parameterization of wave dissipation by sea ice". In: *Cold Regions Science and Technology* 199, p. 103582. ISSN: 0165-232X. DOI: `https://doi.org/10.1016/j.coldregions.2022.103582`.

---

## Author Comment (AC2)

**Response to RC2**

**1 General comments**

**1.1**

This paper describes a method for determining ocean wave induced sea ice breakup patterns using the total bending energy, rather than a local maximum strain criterion. The authors simplify the model by assuming quasi-static bending and by only considering a pseudo one-way coupling from the fluid to the ice deformation. (I say pseudo one-way coupling because the authors still use a sea-ice specific dispersion relation and a model for attenuation by sea ice, although the fluid displacement appears in the equations as a forcing term rather than as an unknown.) Of course, simplifications such as these are necessary, but I would have liked to have seen a little more discussion around these modelling decisions in section 2.1.

You are right that we chose to simplify the model as much as we could to focus on one process: the fracturing and associated temporal evolution of a fracture front. The original paper presented some of the reasoning behind our decisions in the introduction to Sect. 2. This introduction is now transformed into a subsection, slightly expanded, and the introduction section amended to justify our decisions earlier. In particular, a point we had not made clear enough, is that our model is meant to be iterated in time, while a model like that of Mokus and Montiel (2022) can only be iterated between successive steady state, an assumption we meant to relax as we are interested in the progression of a fracture front. Additionally, we added an appendix comparing the results (in terms of curvature and elastic energy) of using the formulation presented here, to that of Mokus and Montiel (2022) (for convenience, as it was developed by the first author and aimed at similar goals). Our results suggest that random fluctuations of the wave state have more influence than changes in ice thickness, significant wave height, or even floe length, when it comes to computing the elastic energy of a deformed floe.

**1.2**

The results section is built around comparison with the experimental data of Auvity et al. (2025). I would have also liked to see a numerical comparison with the critical strain fracture criterion. For instance, by imposing an incident wave, the proposed energy method and the strain criterion method would lead to different breakup patterns, and I am left wondering what the qualitative differences between these might be. I don't think addressing this point is necessary for publication of this paper, but it would strengthen the current paper or be an interesting question to address in a follow up work.

We thank you for this valuable comment, as it helped us clarify why we did not include a comparison of the two criteria in our paper. We indeed do not compare simulations with the critical strain fracture criterion here because in the experiments of Auvity et al. (2025), this critical strain does not seem to exist. Another way of saying this is: it does not come up as a material property, but depends on the wave forcing. We therefore could not prescribe such material property in our model for comparison of the two criteria. The absence of a constant critical strain is a salient point of the work of Auvity et al. (2025), and a motivation for our own work (which we alluded to in our introduction). We remind the readers of our choice of not performing a comparison between the two criteria in the current Sect. 3 (1 278) and Sect. 5 (1 478) of the manuscript.

However, you are right in that this comparison needs to be done. We have started work on establishing a mapping between the two criteria, both by making measurements in the lab and by using our numerical model. Ultimately, we want to establish whether choosing one of the two criteria will impact a modelled floe size distribution, which we mention in the closing sentence of this paper. It is still in early stages, and will hopefully be the subject of a later published study.

**1.3**

With those issues pointed out, I must conclude by saying that the paper addresses an important point in the sea-ice breakup literature with a novel idea. It is very well written and well presented with excellent figures, and I recommend it for publication once the issues raised in this review have been addressed.

We thank you for these kind words and the time you dedicated to reviewing our paper. Below we address your minor comments.

**2 Specific comments**

Some more minor issues are listed below:

**2.1**

Line 41:  $be \rightarrow been$ Corrected.

**2.2**

It should be noted that equation (1) is Archimedes' principle.

Thank you, this precision has been added.

**2.3**

**Is the energy release rate G for ice floes/other materials known or easy to measure?**

This parameter can be measured in the laboratory or on the field, either directly, or from the fracture toughness to which it is related through the Young's modulus, for example by three-point bending tests. Our original manuscript alluded to it in the introduction (line 59), where we now make this point clearer. As it usually is the case, the mechanical properties of sea ice are less well constrained than that of other more standard material, or even fresh water ice, and can be expected to depend on temperature and brine volume fraction, and more generally on the history of the material. Timco and Weeks (2010) compiled previous studies of fracture toughness measurements. Wei and Dai (2021) conducted such measurements more recently, at the lab scale, and compared dry and wet samples. We added Section 2.4.4 to inform the reader on the values this parameter (as well as critical strain) can take.

**2.4**

Figure 2 is very helpful for understanding the fracture process. I would suggest adding a little further discussion about this figure at line 174. For instance: It would be helpful to demystify the algorithmic/procedural steps. E.g. if I understand correctly, for each fracture location, the bending must be computed from (6), before the energies can be calculated.

We thank you for this suggestion. We added a more detailed description of how the fracture search is implemented, as well as a flowchart illustrating it, at the end of section 2.3.1.

**2.5**

Is it correct to say (in a simplified sense) that the right fragment energy is generally decreasing in 2a because the right fragment is becoming shorter.

Yes, it is correct. The situation illustrated in this figure corresponds to the "fully developed" case, which has the analytic solution presented in appendix. The expression of the elastic energy is the sum of fifteen terms (for monochromatic cases) and is not straightforwardly analysed. Intuitively though, it tends to increase with increasing floe length (while oscillating). If wave amplitude is attenuated ( $\alpha \neq 0$ ), it eventually tapers off; otherwise, it keeps growing. For finding fractures, we compute this energy for two hypothetical floes that conserve the length of the original floe. The shorter they are, the smaller their potential energy. If the hypothetical fracture lies close to the left edge of the (original) floe, the left fragment will have a small energy; if it lies close the right edge of the floe, the right fragment will have a small energy. The abscissa of Fig. 2a (original manuscript) being the location of that hypothetical fracture,  $x_1$ , for increasing  $x_1$  the length of the left fragment increases (so does its energy) and the length of the right fragment decreases (so does its energy).

**2.6**

**Line 222: Please define a semi-normal kernel**

The definition is given in Eq. (24) of the original manuscript. We consider a Gaussian function with centre  $\mu$  and width  $\sigma$ ; our kernel is defined piecewise, 1 left of  $\mu$  and Gaussian right of  $\mu$ . We changed the formulation to 'semi-Gaussian' in the revised manuscript, as the Gaussian is not normalised in the way a normal density function would be (so that our kernel is continuous and equal to 1 on both sides of the transition at  $\mu$ ).

**2.7**

When discussing Auvity et al. (2025) in section 3.2, can the authors elaborate on what is meant by the requirement of fracture on

**nonlinear waves? What kind of nonlinearities are they referring to?**

We mean that in the lab, for fracture to occur, the forcing amplitude has to be high relative to the wavelength. This is typically quantified with the wave slope ak, with linear water wave theory classically considered valid for ak < 0.1. The wave profiles observed by Auvity et al. (2025) thus depart from sine waves, and more closely resemble triangular waves close to the fracture onsets. Their measured wave slopes are about 0.14, a precision we added to our manuscript.

**References**

- Auvity, B., L. Duchemin, A. Eddi, and S. Perrard (2025). Wave induced fracture of a sea ice analog. DOI: 10.48550/ARXIV.2501.04824. arXiv: 2501.04824 [physics.flu-dyn].
- Mokus, N. G. A. and F. Montiel (2022). "Wave-triggered breakup in the marginal ice zone generates lognormal floe size distributions: a simulation study". In: *The Cryosphere* 16.10, pp. 4447–4472. DOI: 10.5194/tc-16-4447-2022.
- Timco, G. W. and W. F. Weeks (2010). "A review of the engineering properties of sea ice". In: *Cold Regions Science and Technology* 60.2, pp. 107–129. ISSN: 0165-232X. DOI: https://doi.org/10.1016/j.coldregions. 2009.10.003.
- Wei, M. and F. Dai (Aug. 2021). "Laboratory-scale mixed-mode I/II fracture tests on columnar saline ice". In: *Theoretical and Applied Fracture Mechanics* 114, p. 102982. ISSN: 0167-8442. DOI: 10.1016/j.tafmec.2021. 102982.

---

## Author Comment (AC3)

**Response to RC3**

**1 Opening statement**

In this manuscript, the authors discuss a novel wave-induced sea ice criterion/fracture model. I think this is a generally interesting and well written study, and I am, in general, supportive of publication. I have a few comments that I would like the authors to consider and I think that the manuscript should be published once these are addressed.

We thank you for taking the time to review our manuscript and for your valuable comments, which we answer point-by-point below.

**2 General comments**

**2.1**

Regarding section 2.1: while this is definitely interesting, I wonder how big the effect of solving the ice motion, not just assuming that the ice follows the waves (which I agree is strictly speaking incorrect), is. If I understand correctly, the results from 2.1 are used in all the following? I would be curious to see (either as additional lines in some plots, or as an appendix), a quick analysis/comparison of how much difference there is between the results using the 'floes following the water' vs. the 'floes moving following a balance between buoyancy and flexure' approximations. Is this a large meaningful difference in 'standard' waves in ice swell conditions, or just a minor 'distraction'?

The formalism we expose in Sect. 2.1 is, indeed, use throughout. However, it should noted that Sect. 2.2 is largely independent from it. The focus of the current paper is a fracturing process and associated criterion. To inform it, we need the potential energy of the floe, derived from its curvature. In Sect. 2.2,

we merely offer a way to access this curvature through a linear, elastic plate model, in a way that can be time-stepped (a very important feature, as we intent to study with our model time-dependent fracture-related processes such as the propagation of fracture fronts) and is computationally efficient; but any other sensible way to input a curvature in our fracturing criteria could be substituted, after a dedicated implementation. In particular, measured curvatures could be used. We agree that this was not stated clearly enough in our original submission, and we have altered the manuscript to convey our meaning better by expanding Sect. 2.4.2. Additionally, we added an appendix comparing the results (in terms of curvature and elastic energy) of using the formulation presented here (ice not conforming to the sea surface), to that of Mokus and Montiel (2022) (ice conforming to the sea surface, for convenience, as it was developed by the first author and aimed at similar goals). Our results show that random fluctuations of the wave state have more influence than changes in ice thickness, significant wave height, or even floe length, when it comes to computing the elastic energy of a deformed floe.

**2.2**

I don't have any major concerns about the results presented from a 'mathematical' point of view. However, this field of study has had (in my opinion, but this may be controversial) a history of offering 'mathematically rigorous' explanations and models that may have actually turned out to be 'physically wrong' because in the real world, a different physical mechanism dominates. Since we are in a branch of applied physics, the ground truth we should compare to is field data, and a model per se, independently of its elegance and mathematical correctness, has no real value unless it explains the real world data better that similar or higher complexity competing models. I understand that the authors compare their results to idealized experiments, but there are so many issues with scaling, ice conditions and formation and structure, etc, that in my experience experiments often have a limited power of proof in this field—for example, the relative scaling between mechanisms and the dominating physics may be different between the field and the laboratory. In this regard, I would like to see more discussion about the following:

**2.2.1**

Though I understand this may be discussed in the reference provided, I believe that an in-depth discussion of the experimental conditions, ice conditions, etc, from Auvity et al, would be useful, being 'self critical/self skeptical' and making it clear what the possible limitations are, would be useful.

We think there is some confusion about the ice conditions in the experiment of Auvity et al. As explained in the beginning of Sect. 3 (Numerical experiment), which outlines the physical experimental setup, and mentioned in the Introduction, Results, and Conclusion sections, the material considered is not ice, but a mechanical analogue of ice (a varnish layer). Again, the goal of the comparison conducted here between the model and experiment is to place ourselves in conditions in which we can focus on and understand one physical mechanism, the fracture of a brittle solid by waves, and evaluate a fracturing criterion. Therefore, even though this material is not ice, the point is that it is a brittle continuous solid, with mechanical parameters that are comparable to ice. It presents the advantage to be solid at room temperature and easier to control experimentally in terms of thickness and homogeneity. Our Table 1 sums up these conditions by providing all necessary numerical values. Comparisons with similar experiments performed on ice are underway.

In terms of the possible limitations of our comparison, we had dedicated Sect. 3.2 to discussing perhaps the biggest one (the linearity of the waves) and discussed a second (layering of the varnish) in the Discussion section. We kindly ask you to refer to Auvity et al. (2025) for details not present in our study, as our goal was to present the necessary information while keep the length of our paper reasonable.

Again, we specifically chose this experiment for a basis of comparison because it is simple (it does not entail all of the variations in sea ice formation, conditions, etc. that an in situ experiment could), and therefore it allows us to isolate one physical (not mathematical) effect, that of brittle fracture. We hope our revised manuscript communicates these better.

**2.2.2**

Can you present simple scaling analysis between your experimental data and typical field conditions, focusing on non dimensional groups that are relevant/appear in your model? Do this seem to scale the same (in which case, one can reasonably hope that the present model may be transferable to real world field data if there

are no surprises (other physics) happening), or do these have large mismatches (in which case, there would still be a significant burden of proof)? Compiling all of this in a discussion and dedicated table would be useful.

Thank you for this interesting comment. We indeed spent quite a lot of time reflecting on non-dimensional quantities. The one quantity we have identified, is the dimensionless wavenumber  $kL_D$ . We use it as an explanatory (that is, independent) variable; it does not, in itself, control breakup but emerges naturally from the bending ODE. It is noteworthy that this quantity seems to exist on a very narrow range: Auvity et al. (2025) obtained fracture for  $0.05

Figure 1: Distribution of dimensionless wavenumbers, determined from estimating wave and ice properties following Voermans et al. (2020), and sampling from the triangular distributions they suggest. The boxplots are colour-coded based on whether the study observed breakup or not for these properties. The whiskers are defined as 1.5 times the interquartile range in log units.

a neat mathematical exercise, I would need to see a comparison to field data. I see two possibilities here: either 1 use existing already processed field data, directly from some of the references provided (for example Voermans et al that is referred there and may have been relying on open data/provide enough data to ensure reproducibility of the results, or some of the other references), or 2 perform your own such comparison with your own methodology from scratch based on data you have gathered, or that are publicly available. However, I understand that this is possibly a significant amount of work (maybe not for 1 if there is a smart way to reuse the data analysis previously performed, but definitely with 2), so I don't really feel that I can 'require' the authors to do so in this paper. Still, I think that the authors should either go through the (possibly significant amount of work) task of doing such a comparison, or if not, at least have a very clear discussion about the fact that there is still a significant 'burden of proof' on the present method to demonstrate its applicability to real world data, putting more weight on the possible limitations of the present model, and suggesting how to test this model.

It seems we might have different points of view on what is mathematical and physical. We consider that our model is 'mathematical' in the sense that it is a closed set of equations programmed into a computer. But we try to approach a physical problem, with a 'physical' parametrisation of the breakup process based on Griffith's fracture theory and a 'physical' constitutive relation. Yes, at the end of the day, the model is 'wrong' but still might be useful. For us, the most logical thing to do to 'try' it was to compare it to a laboratory experiment also designed to study in isolation a single process, that of brittle fracture of an elastic solid by waves. The laboratory setting has the advantage of being a much more controlled environment; the observation of wave-induced ice breakup being serendipitous in nature, and uncertainty on field measurements being what they are.

We would like to point out Voermans et al. did not provide processed data, but provided their buoys data. Most of the data, in particular pertaining to the ice properties, necessary to compute their fracture threshold was estimated from empirical relations. Additionally, their breakup criterion can be rewritten as the scaled ratio of the maximum curvature undergone by the ice to a critical curvature; a quantity that, again, we do not have access to in the case of the analogue material, for we believe it does not 'exist', that is, it is a function of the wave forcing.

Finally, we are in the process of evaluating our model on data we participated in acquiring (Kuchly et al. 2025). We have amended our discussion to highlight the present study is (beyond the presentation of the model itself) a first step towards validating our formalism, and what our intentions are for the future.

**2.2.4**

The authors discuss quite a bit previous, existing parameterization methods in the introduction. I would like to see this thread picked up more in the results and discussion, and ideally a comparison of both the mathematical behavior of these pre existing parameterizations vs. your present model (typical scaling—is it the same? different? in scaling itself, or prefactors?), and possibly theory prediction power comparisons. Are you predictions significantly different from previous parameterizations? If not, what is the added value of your model? If yes, given that in particular the 'empirical criterion based' methods seem to do a reasonable work at fitting observations with ad hoc tuning, do you trust that your model is right and previous parameterizations fitted on field data are wrong / how do you do better with your present model than the previous fitted parameterizations?

One limitation of the previous method we discuss in our paper, the maximum strain criterion, is that it requires an extra parametrisation. We mention this point in our introduction. Our approach does not require this extra parametrisation, which we consider a plus, and is grounded in fracture mechanics. We believe however that the 'good' approach would be to identify what quantities are actual material properties (intensive physical properties, which do not depend on the geometry or the size of a floe), with eventual dependencies to environmental factors such as temperature or brine fraction, but not to the kind of forcing (typically, in a linear regime and ignoring fatigue, we expect no dependency to the wavenumber). The Young's modulus, the energy release rate should be among these. The critical strain does not seem to be one, at least when studying a physical analogue, which was chosen because it also behaves as a brittle elastic solid. As mentioned in our response to your previous comment, experiments on ice are ongoing and will be used to determine whether the absence of a constant critical strain can be assumed for ice, as well. We explicitly added this mention to our discussion section.

In a sense, we can consider that our integrated, energy fracture criterion is more general than a parametrisation based on maximum strain. It may be however that the two are equivalent in some conditions and with respect to some metrics such as the size distribution of fragments, or the speed of the fracture front; evaluating this would require some sort of mapping between fracture energy and critical strain (the values parametrising either formulation). It is likely this mapping will also involve the forcing wave, in some fashion. We are in the process of deriving such a mapping (which we alluded to in the closing paragraph of our manuscript) and conducting these experiments, the results of which will hopefully be part of a future publication. If we establish the parametrisations are equivalent, of course, it will be sensible to stick to the simplest, or 'numerically cheapest', one. Our postulate is that for now, we do not know, but we note that a constant strain threshold independent of wave conditions does not explain existing experimental results or our own results.

Finally, the other advantage of our model, that we had not made explicitly clear in our original submission, is that it can be stepped in time in a rational way, which allows for not only looking at the final state of a breakup event (for example, Mokus and Montiel 2022), but at the transient evolution of a breakup front: its speed, sizes of initial fragments, secondary fracture, etc. We hope we have made this point clearer in our revised manuscript.

**References**

- Auvity, B., L. Duchemin, A. Eddi, and S. Perrard (2025). Wave induced fracture of a sea ice analog. DOI: 10.48550/ARXIV.2501.04824. arXiv: 2501.04824 [physics.flu-dyn].
- Dolatshah, A., F. Nelli, L. G. Bennetts, A. Alberello, M. H. Meylan, J. P. Monty, and A. Toffoli (Sept. 2018). "Letter: Hydroelastic interactions between water waves and floating freshwater ice". In: *Physics of Fluids* 30.9, p. 091702. ISSN: 1070-6631. DOI: 10.1063/1.5050262.
- Herman, A. (2018). "Wave-Induced Surge Motion and Collisions of Sea Ice Floes: Finite-Floe-Size Effects". In: *Journal of Geophysical Research: Oceans* 123.10, pp. 7472–7494.
- Kuchly, S., B. Auvity, N. G. A. Mokus, M. Bureau, P. Nicot, A. Fourgeaud, V. Dansereau, A. Eddi, S. Perrard, D. Dumont, and L. Moreau (2025).
  "An integrated multi-instrument methodology for studying marginal ice zone dynamics and wave-ice interactions". In: EGUsphere 2025, pp. 1–23.
  DOI: 10.5194/egusphere-2025-3304.
- Mokus, N. G. A. and F. Montiel (2022). "Wave-triggered breakup in the marginal ice zone generates lognormal floe size distributions: a simulation study". In: *The Cryosphere* 16.10, pp. 4447–4472. DOI: 10.5194/tc-16-4447-2022.

- Passerotti, G., L. G. Bennetts, F. von Bock und Polach, A. Alberello, O. Puolakka, A. Dolatshah, J. Monbaliu, and A. Toffoli (2022). "Interactions between irregular wave fields and sea ice: A physical model for wave attenuation and ice breakup in an ice tank". In: *Journal of Physical Oceanography* 52.7, pp. 1431–1446.
- Voermans, J. J., J. Rabault, K. Filchuk, I. Ryzhov, P. Heil, A. Marchenko, C. O. Collins III, M. Dabboor, G. Sutherland, and A. V. Babanin (2020). "Experimental evidence for a universal threshold characterizing wave-induced sea ice break-up". In: *The Cryosphere* 14.11, pp. 4265–4278. DOI: https://doi.org/10.5194/tc-14-4265-2020.

---

## Author Response (AR2)

Dear Qiang,

Please find attached our revised manuscript and the difference file.
Responses to the reviewers' comments can be found concatenated after this letter.

Best regards,
Nicolas Mokus

**Response to RC1's report**

*The already good manuscript has been improved following Reviewers' suggestions. I personally enjoyed the addition of Sec 2.6. I will leave here few minor comments for potential consideration by the authors.*

**1 Minor comments**

**1.1**

*It would be great if SWIIFT will find its way in coupled ocean-sea ice models in which waves are also accounted for. For this to be feasible, the computational time of the SWIIFT module should be small. Within this context, it would be appreciated a comment on the computational cost of the algorithm, e.g. what is the CPU time for the example in Sec 2.6? What is the computational cost of SWIIFT relative for example to Swell fracture in Horvat and Tziperman 2015?*

Thank you for this relevant comment. If our ultimate goal would indeed be for our results to percolate into global climate models, we are not for the moment thinking of running SWIIFT alongside (as a module) these large scale models. We designed it primarily as a tool to get a better grasp of a physical process and derive associated parametrisations, in particular at a fine scale (that of floes) and when it comes to time evolution (something Horvat and Tziperman (2015) cannot do, as it considers instant fracture along a full grid cell). For instance, we can use SWIIFT to follow the progress of the breakup front. Comparison with observations of this process is our current goal.

Because SWIIFT resolve time-evolution and fine spatial scales, it is unfortunately slow. It is also consequence of, on the first hand, the sought convergence in fracture patterns which requires a short timestep; and, on the second hand, the systematic approach to fracture lookup, which requires a

first discretisation of the hypothetical fracture locations (with a resolution high-enough to detect local maxima and avoid aliasing of the expected free energy) and as many local minimisations as pairs of local maxima were found. It can be accelerated through several axes:

- Tweaking the termination condition of the minimisation algorithm. The default condition of the SciPy function we use is $1 \times 10^{-5}$, which means we seek convergence in floe lengths of the order of $10\,\mu\mathrm{m}$, which is quite unnecessary.

- Moving from local to global optimisations. Several methods exist, either stochastic or deterministic. The limitation of either is that finding the true global minima is never guaranteed, which can be an acceptable tradeoff when running an ensemble of simulations with added noise.

- Parallelising the fracture search. This can be done either by parallelising the local minima search on a single floe, or parallelising fracture search across several floes.

The first point is trivial beyond some eventual sensibility testing, and we have started to work on the second point. As illustration, the first $20\,\mathrm{s}$ of the simulation presented in Sect. 2.6 take 41 minutes to run; the first $9\,\mathrm{s}$ (1050 timesteps) of these take only thirty-one seconds, as the conditions for fracture are not met (the wavefield is not energetic enough for the elastic energy of the initial floe to exceed the fracture energy) and, as a result, no fracture search is conducted.

However, when it comes to deriving parametrisations for the coupling of our wave–ice interaction model to large scale models, an obvious method we are considering would the preliminary building of a table predicting some floe size distribution statistics out of wave and ice conditions, as in Roach et al. (2018). These were then used to train neural networks to considerably reduce the associated overhead (Horvat and Roach 2022).

**1.2**

*In the intro, where the MIZ response to storms is discussed, I leave a couple of references that the authors can consider adding (Vichi, 2022, TC, https://tc.copernicus.org/articles/16/4087/2022/tc-16-4087-2022.html; Cavallo et al, Comms Earth Env, https://www.nature.com/articles/s43247-025-02022-9).*

Thank you for these suggestions. We added the reference to Cavallo et al. We appreciate the other reference, but we think its topic (identifying

the marginal ice zone from satellite products) is not directly implementable into our discussion and would require introducing the broader context, which would lengthen the text.

**1.3**

*2.4.2 "Wave state" sounds unusual, the preferred forms are "sea state" or "wave properties".*

Thank you for the suggestion. We changed the Section title to "Sea state".

**1.4**

*2.6 L351, I think a better and clearer phrasing would be "wave height $H_S = 0.5\,\mathrm{m}$ and peak period $3.84\,\mathrm{s}$, ...", since the $H_S$ and peak period can be prescribed in dependently of each other.*

Thank you for this comment. In their seminal paper, Pierson and Moskowitz introduced the Pierson–Moskowitz spectrum as depending on one parameter, the $19.5\,\mathrm{m}$ wind speed as measured by a given ship. Modern formulations used by various wave modellers express the spectrum parametrised by the significant wave height *or* the peak frequency (or the wind speed). There is thus a mapping between significant wave height and peak period, that can be expressed

$$\frac{\alpha g^2}{(2\pi)^4} = \frac{5}{16}\frac{H_{\mathrm{S}}{}^2}{T_p{}^4} \tag{1}$$

with $\alpha = 8.1 \times 10^{-3}$, following the guidelines of the International Towing Tank Conference (Stansberg et al. 2002).

This spectrum is generalised into the Bretschneider family, with two independent parameters. Unfortunately, there is a lack of precision in the literature when discussing these spectral formulations, so that the terms Bretschneider spectrum or two-parameter Pierson–Moskowitz spectrum may be used indiscriminately by different authors. Herein, we specifically used the one-parameter Pierson–Moskowitz spectrum Our phrasing thus convey our intended meaning: setting the significant wave height unambiguously sets the peak period. We added this precision to our manuscript.

**References**

Cavallo, S. M., M. C. Frank, and C. M. Bitz (Jan. 2025). "Sea ice loss in association with Arctic cyclones". en. In: *Communications Earth & Environment* 6.1. DOI: 10.1038/s43247-025-02022-9.

Horvat, C. and E. Tziperman (2015). "A prognostic model of the sea-ice floe size and thickness distribution". In: *The Cryosphere* 9.6, pp. 2119–2134. DOI: 10.5194/tc-9-2119-2015.

Horvat, C. and L. A. Roach (Jan. 2022). "WIFF1.0: a hybrid machine-learning-based parameterization of wave-induced sea ice floe fracture". In: *Geoscientific Model Development* 15.2, pp. 803–814. ISSN: 1991-9603. DOI: 10.5194/gmd-15-803-2022.

Pierson, W. J. and L. Moskowitz (1964). "A proposed spectral form for fully developed wind seas based on the similarity theory of SA Kitaigorodskii". In: *Journal of geophysical research* 69.24, pp. 5181–5190. DOI: https://doi.org/10.1029/JZ069i024p05181.

Roach, L. A., C. Horvat, S. M. Dean, and C. M. Bitz (2018). "An emergent sea ice floe size distribution in a global coupled ocean-sea ice model". In: *Journal of Geophysical Research: Oceans* 123.6, pp. 4322–4337.

Stansberg, C. T., G. Contento, S. Hong, M. Irani, S. Ishida, and R. Mercier (Jan. 2002). "The specialist committee on waves: Final report and recommendations to the 23rd ITTC". In: pp. 505–736.

**Response to RC3's report**

*I want to thank the authors for their detailed answers. As stated in the previous review round, I am curious of how applicable this model will be to real world data in a broad range of conditions. I think that the discussion about what is mathematically vs. physically correct (not just at the scale of the laboratory or small scale experiments but also at the full scale of field realistic conditions), which we have had through this review round, is not simple and not settled out by the present work. But at the same time, I am aware that an exhaustive study comparing this model to field data would be too much work for including in this paper. Therefore, I think that the present work can be accepted as is: I believe it is strictly speaking mathematically correct given the assumptions that are made, as demonstrated by validation against idealized conditions in the laboratory, and I think it is acceptable to have it published as an attempt at modeling which mathematical correctness is validated in controlled conditions, with in-depth comparison and validation against full scale real world data planned for the future. I will follow developments in this regard with great interest, and I hope the authors can present such a work in the future, as, in my experience, the field of research about waves in ice attenuation is a good illustration that a model can be mathematically (and even physically) correct at one scale (for example the laboratory), but hard to transfer to the full scale due to changes in scaling and changes in the dominant physics. However, I am willing to recognize that this view is mine, and is likely quite heterodox, so I think that the authors should get the possibility to publish their findings and results, and get the possibility to check in the future if and how well this model applies also to full scale data.*

We thank you for your understanding. We appreciate that the laboratory scale cannot necessarily be one-to-one transformed to field scale, due to numerous added complexities at the field scale (spatial heterogeneities, additional feedback processes between the ice, atmosphere and ocean, and so

on). We believe an important step before addressing these complexities was to validate the model against a simple experiment, allowing to isolate specific physical processes and their model representation (here, the fracturing). However, even though it lacks formal validation, we believe for now the example of spectrally-forced, time-dependent fracture showed in the new Sect. 2.6 is a good illustration that our model can be run at a more geophysical scale. We now strive to conduct comparisons to field data.